# An explicit estimate of the atmospheric nutrient impact on global oceanic productivity

Stelios Myriokefalitakis[1], Matthias Gröger[2], Jenny Hieronymus[3] and Ralf Döscher[3]

[1] Institute for Environmental Research and Sustainable Development (IERSD), National Observatory of Athens, Penteli, Greece
[2] Leibniz Institute for Baltic Sea Research Warnemünde (IOW), Rostock, Germany
[3] Swedish Meteorological and Hydrological Institute (SMHI), Norrköping, Sweden

*Correspondence to*: Stelios Myriokefalitakis (steliosm@noa.gr) and Matthias Gröger (matthias.groeger@io-warnemuende.de)

**Abstract.** State-of-the-art global nutrient deposition fields are here coupled to the biogeochemistry model PISCES to investigate the effect on ocean biogeochemistry in the context of atmospheric forcings for preindustrial, present, and future periods. PISCES, as part of the EC-Earth model suite, runs in offline mode using prescribed dynamical fields as simulated by the ocean model NEMO. Present-day atmospheric deposition fluxes of inorganic N, Fe, and P into the global ocean are accounted equal to ~40 Tg-N yr$^{-1}$, ~0.28 Tg-Fe yr$^{-1}$, and ~0.10 Tg-P yr$^{-1}$. Preindustrial atmospheric nutrient deposition fluxes are lower compared to present-day (~51%, ~36%, and ~40% for N, Fe, and P, respectively). However, the impact on global productivity is low (~3%) since a large part of marine productivity is driven by nutrients recycled in the upper ocean layer or other local factors. Prominent changes are, nevertheless, found for regional productivity. Up to 20% reductions occur in oligotrophic regions such as the subtropical gyres in the northern hemisphere under preindustrial conditions. In the subpolar Pacific, reduced preindustrial Fe fluxes lead to a substantial decline of siliceous diatom production and subsequent accumulation of Si, P, and N, in the subpolar gyre. Further southward transport of these nutrient-enriched waters leads to strongly elevated production of calcareous nanophytoplankton further south and southeast, where iron no longer limits productivity. The North Pacific is found most sensitive to variations in depositional fluxes, mainly because the water exchange with nutrient-rich polar waters is hampered by land bridges. By contrast, large amounts of unutilized nutrients are advected equatorward in the Southern Ocean and North Atlantic making these regions less sensitive to external nutrient input. Despite the lower aerosol N:P ratios with respect to the Redfield ratio during the preindustrial period, the nitrogen fixation decreased in the subtropical gyres mainly due to diminished iron supply. The projected future changes in air pollutants under the RCP8.5 emission scenario result in a modest decrease of the atmospheric nutrients inputs into the global ocean compared to present-day (~13%, ~14%, and ~20% for N, Fe, and P, respectively), without significantly affecting the projected primary production in the model. Sensitivity simulations further show that the impact of atmospheric organic nutrients on the global oceanic productivity turns out roughly as high as the present-day productivity increase since preindustrial times when only inorganic nutrients' supply is taken into account in the model. On the other hand, variations in atmospheric phosphorus supply have almost no effect on productivity in the model.

# 1 Introduction

Marine primary production is a critical component of the global carbon cycle and important for sustaining the habitability on Earth, although vulnerable to environmental changes (e.g., Steinacher et al., 2010). For example, an estimated decline of subarctic productivity has been reported to accompany the warming of the last 150 years (Osman et al., 2019). Global warming induced by greenhouse gas emissions has increased ocean stratification, reducing the supply of nutrients from subsurface waters and inhibiting the growth of phytoplankton in the surface ocean (Behrenfeld et al., 2006). Thus, the role of nutrient supply by atmospheric deposition will likely be more important in a warmer climate. Several studies have documented the importance of primary production on the surface ocean $CO_2$ concentrations (e.g., Falkowski et al., 2000; Gruber, 2004; Gruber et al., 2009; Le Quéré et al., 2013; Smith, 2019) via carbon uptake and sinking of the particulate organic matter to the deeper ocean. However, significant uncertainties remain in the projected production among state-of-the-art model simulations, which can range between 2 and 20% for the coupled model intercomparison project phase 3 (CMIP3) and CMIP5 models (Fu et al., 2016; Steinacher et al., 2010), mainly due to the different responses of phytoplankton production to changes in water temperature and stratification (Gröger et al., 2013; Laufkötter et al., 2016; Steinacher et al., 2010).

During primary production, the growth of the phytoplankton functional types (e.g., diatoms and nanophytoplankton) results in a newly formed particulate organic matter within the euphotic zone. These processes are limited however by light, temperature, and nutrients' availability. Nutrient inputs to the euphotic upper ocean result from internal ocean dynamics, such as upwelling or external sources, i.e., input by rivers and atmospheric deposition. Effects of riverine inputs have been widely investigated and are mostly restricted along the coasts or in marginal shelf basins (e.g., Behrenfeld et al., 2006; Gröger et al., 2013; Holt et al., 2012). Hence, the atmospheric deposition is the only external supply that can reach distal open ocean regions far remote from land.

Human activities have heavily perturbed the atmospheric chemical composition (e.g., Mahowald et al., 2017), but the impact of atmospheric composition on marine biogeochemistry, and consequently on the oceanic carbon- and nutrient cycles, is rather complex and still not fully understood. Among other species deposited into the open ocean, nitrogen (N), iron (Fe), phosphorus (P), and silicon (Si) are the nutrients that significantly limit the marine phytoplankton growth rates and thus directly impact on ocean-atmosphere carbon fluxes, in particular where nutrients are the growth-limiting factor for phytoplankton.

Atmospheric nitrogen inputs to the global ocean are mainly derived from anthropogenic combustion and agricultural sources over densely populated regions (Duce et al., 2008). So far, it is widely accepted that the marine biota primarily utilizes the inorganic nitrogen in its oxidized form (i.e., nitrogen oxides (NOx), nitric acid ($HNO_3$) and particulate nitrate ($NO_3^-$)), as well as in its reduced form (i.e., ammonia ($NH_3$) and particulate ammonium ($NH_4^+$)) (e.g., Duce et al., 1991). However, there is evidence that the dissolved organic nitrogen (DON) inputs (e.g., from rivers along the coasts) can likewise be also efficiently utilized (e.g., Aumont et al., 2015). In the atmosphere, the global organic nitrogen (ON) cycle has been demonstrated to have a strong (~45%) anthropogenic component (Kanakidou et al., 2012). Kanakidou et al. (2016) calculated that 20%–25% of the

nitrogen deposition is in the form of ON, overall with DON deposition to be about 25% of the total dissolved nitrogen deposition to the global ocean.

Present-day atmospheric nitrogen input to the oceans is estimated to be roughly 39-68 Tg-N yr$^{-1}$ (e.g., Duce et al., 2008; Kanakidou et al., 2016; Krishnamurthy et al., 2007; Wang et al., 2019), and the global oxidized and reduced nitrogen flux has increased from preindustrial values of ~13 Tg-N yr$^{-1}$ to about 40 Tg-N yr$^{-1}$ in modern times (Kanakidou et al., 2016). Moreover, the aforementioned studies suggested that nearly half of nitrogen emissions are transformed into aerosols in the atmosphere, with the nitrogen-containing aerosols having increased by approximately 2.5 times since preindustrial times. This resulted in a doubling of atmospheric soluble N deposition in the ocean due to human activities alone. Atmospheric nitrogen deposition may also significantly impact on the surface water inorganic N/P ratios and thus further influence global nitrogen fixation rates (Moore and Doney, 2007), i.e., the reduction of gaseous N$_2$ to ammonium as catalyzed by nitrogenase. Krishnamurthy et al. (2007) demonstrated that compared to preindustrial conditions, present-day inorganic nitrogen inputs to the oceans from anthropogenic sources could so far be partially compensated by the decreased nitrogen fixation, resulting thus in a modest effect on primary production. For future conditions, however, Wrightson and Tagliabue (2020) showed that the CMIP5 models project a general decrease in N$_2$-fixation by diazotrophs under the Representative Concentration Pathway (RCP) 8.5 emission scenario, although the complexity in response to climate change at a regional scale among the different models.

The most important atmospheric source for marine nutrients (such as Fe, P, and Si) in the open ocean is the mineral dust deposition. Dust aerosols are usually subject to intensive atmospheric processing during their long-range transport over remote oceanic regions. Changes in the properties of mineral aerosols during atmospheric transport involve chemical interactions with air masses (i.e., aerosol aging) that lead to different coatings of dust particles by sulfate (SO$_4^{2-}$), NH$_4^+$, NO$_3^-$, and organics (e.g., oxalic acid). In particular, pollutants from strong acids, such as the sulfuric (H$_2$SO$_4$) and nitric (HNO$_3$) acids, that coat deliquesced minerals, potentially transform part of their contained insoluble forms (e.g., hematite, apatite) into soluble forms (e.g., Fe(II), Fe(III), PO$_4^{-3}$) during atmospheric processing (e.g., Nenes et al., 2011; Shi et al., 2011). This process is further enhanced in the presence of organics such as the oxalic acid, that converts part of the insoluble Fe-containing minerals to soluble organic Fe-complexes, under favored atmospheric conditions (e.g., Paris et al., 2011; Paris and Desboeufs, 2013). However, according to future emission scenarios (van Vuuren et al., 2011), the sources of the main acidic atmospheric species such as the nitrogen oxides (NOx) and sulfur (SOx) are expected to decrease by 34–59% and 75–88%, respectively, from 2010 to 2100, but ammonia (NH$_3$) will increase by 3–55%. This heterogeneity in the projection of acidic and alkaline emissions is expected to non-linearly perturb the atmospheric aerosol acidity (Weber et al., 2016), making overall the atmosphere-ocean interactions even more complex.

Iron is utilized by marine phytoplankton mainly in its dissolved form, although the actual bioavailability may substantially differ from the soluble forms of the deposited nutrients into the ocean (Meskhidze et al., 2019). For example, Rubin et al. (2011) showed that in the tropical and subtropical Atlantic Ocean, some marine organisms (e.g., the *Trichodesmium*) can directly access the mineral (insoluble) particulate iron. Dissolved Fe (DFe) from anthropogenic combustion and biomass burning processes can contribute significantly to the atmospheric inputs into the ocean (e.g., Barkley et al., 2019; Hamilton et

al., 2020; Matsui et al., 2018). However, the aerosols from natural and combustion sources tend to be deposited in different regions of the oceans. For example, the subtropical North Atlantic Ocean and the Arabian Sea receive the majority of Fe originated from natural dust aerosols, in contrast to the Pacific and Southern oceans where the Fe-containing combustion aerosols play a more important role compared to atmospheric dust (Ito et al., 2019b).

Present-day global atmospheric DFe and dissolved P (DP) deposition fluxes into the ocean are calculated in the range 0.2–0.4 Tg-Fe yr$^{-1}$ (Ito et al., 2019a; Myriokefalitakis et al., 2018) and 0.10-0.17 Tg-P yr$^{-1}$ (Mahowald et al., 2008; Myriokefalitakis et al., 2016), respectively. Myriokefalitakis et al. (2016) presented a comprehensive description of the organic forms of P (i.e., soil's organic matter, OP associated with anthropogenic combustion and biomass burning emissions, as well as from primary biological aerosol particles). They also demonstrated that DOP can contribute to DP more than 50% over the equatorial oceanic

regions. Compared to the present day, DP and DFe emissions may have increased by roughly 3 and 6 times respectively, since 1850 (Myriokefalitakis et al., 2015, 2016). Wang et al. (2014) further showed that DP emissions increased due to the extended use of biofuels in the energy production sector in developing countries as well as emissions due to extensive deforestation in South America and Southeast Asia. By contrast, a significant increase for DFe emissions before the 1990s due to coal combustion is demonstrated, followed-up by a decline due to the implementation of air-pollution abatements (Wang et al.,

2015b).

The primary production is strongly linked to nitrogen and iron acquisitions by marine biota. However, surface oceanic nutrient concentrations are strongly impacted by atmospheric deposition on a regional scale. Nitrogen atmospheric inputs were shown to have a significant effect on marine productivity, export production, and carbon uptake in Low-Nutrient-Low-Chlorophyll (LNLC) regions. The impact of N and P atmospheric deposition in strong oligotrophic regions, such as the Eastern

Mediterranean, may lead to an increase in present-day primary production by up to 35% (Christodoulaki et al., 2013), resulting overall in a total phytoplanktonic biomass increase by 16% since preindustrial times (Christodoulaki et al., 2016). Marine global primary production rates are currently estimated in the order of 44 - 67 Pg-C yr$^{-1}$, based on biogeochemistry calculations and satellite-based estimates (e.g., Aumont et al., 2015; Behrenfeld et al., 2005; Gröger et al., 2013; Krishnamurthy et al., 2007, 2009, 2010; Steinacher et al., 2010). Krishnamurthy et al. (2009) suggested that the simultaneous anthropogenic N and

Fe deposition can increase oceanic productivity by 1.5 Pg-C yr$^{-1}$, corresponding overall to a reduction of atmospheric pCO$_2$ level of 2.2 ppm by the year 2100. The impact of soluble iron deposition on the carbon export efficiency, although highly uncertain, indicates overall that the Southern Ocean is more sensitive in the projected fire emissions changes compared to other oceanic regions (Hamilton et al., 2020). Assuming, however, a complete assimilation of anthropogenic nitrogen by carbon fixation, a new marine biological production up to 0.3 Pg-C yr$^{-1}$ could be supported (Duce et al., 2008). Although P deposition

may account for only a small fraction of production (Krishnamurthy et al., 2010), an increase in Fe and P deposition can further enhance the N$_2$-fixation in LNLC oceans (Mahowald, 2011; Moore et al., 2013b).

A large portion of the global ocean, especially the subtropical gyres, is depleted in nitrate and phosphate, and consequently, sustain low productivity (e.g., Moore et al., 2013a). About 40% of the global ocean is estimated to be N-limited (Krishnamurthy et al., 2009; Wang et al., 2015a), with most of the remaining to be Fe-limited. The relative larger increases in N than P

deposition in many oceanic regions of the globe cause shifts from N to P limitation (Krishnamurthy et al., 2009; Moore et al., 2013a). On the other hand, many studies suggest that anthropogenic Fe deposition is the most important factor for carbon uptake (Krishnamurthy et al., 2010; Okin et al., 2011), mainly due to its positive effect on productivity in High-Nutrient-Low-Chlorophyll (HNLC) regions. Accordingly, the essential role of iron in oceanic productivity is well established (Tagliabue et

al., 2017) and currently routinely included in marine biogeochemistry models (Aumont and Bopp, 2006; Hajima et al., 2019; Hamilton et al., 2020; Ito et al., 2019b; Moore et al., 2001; Tagliabue et al., 2014, 2016). A further effect of iron is that it can stimulate nitrogen fixation (Camarero and Catalan, 2012; Schulz et al., 2012) because $N_2$-fixing species (diazotrophs) have elevated Fe requirements (Kustka et al., 2002). For example, the $N_2$-fixation is found to be suppressed due to iron-limitations in the eastern tropical Pacific (Wang et al., 2019). Hamilton et al. (2020) further demonstrated that changes in iron deposition

fluxes into the global ocean may affect up to 70% the marine nitrogen cycle via increases in denitrification and nitrogen fixation rates. Present-day global nitrogen fixation is currently estimated in the range of ~111-163 Tg-N yr$^{-1}$ (e.g., Aumont et al., 2015; Krishnamurthy et al., 2009; Wang et al., 2019). Wang et al. (2019) suggested that roughly half of the export production in the subtropical gyres is due to the nitrogen from microbial fixation and external inputs, such as rivers and atmospheric deposition. Regarding the atmospheric inputs, however, the aforementioned study demonstrated based on inversion calculations a potential

decrease (~10%) of $N_2$-fixation, as a response to the elevated present-day nitrogen emissions.

The present study aims to analyze the impact of a comprehensive representation of atmospheric inputs to oceanic productivity. For this, a state-of-the-art marine biogeochemical model is used to integrate the recent knowledge of the atmospheric nutrient deposition fluxes into the ocean, driven by natural and combustion emissions, along with further processing during atmospheric transport. The outlined variable composition and varying sources of the deposited nutrients (i.e., N, Fe, and P) used in this

work, have been recently modeled with a state-of-the-art atmospheric chemistry and transport model based on preindustrial, present, and future emissions. The description of the biogeochemical model and the parameterizations used in the atmospheric chemistry transport model, which determines the atmospheric deposition fields of this work, are presented in Sect. 2. A detailed description of the regional changes in deposition fluxes and the linked atmospheric processes controlling them is also provided. In Sect. 3, the modeled nutrient oceanic concentrations are presented, and the relevant biogeochemical processes, such as the

nitrogen fixation and the primary production, are discussed and also compared to estimates from observations and other modeling studies. The role of present-day air pollutants on nutrients' atmospheric deposition is also discussed, via the comparison of experiments that are forced from atmospheric inputs of preindustrial and projected anthropogenic and biomass burning emission scenarios. The impact of the atmospheric nutrients' organic fraction on the global oceanic productivity is assessed in Sect. 4. Moreover, the implications of our findings concerning the above biogeochemistry parameters are discussed

in Sect. 5. Finally, the main conclusions are summarized in Sect. 6, together with proposed directions for future marine biogeochemistry studies.

## 2 Model description

The state-of-the-art biogeochemistry model PISCES (Aumont et al., 2015), enabled within the framework of the European Community Earth System Model EC-Earth (http://www.ec-earth.org/), is here used in offline modus to investigate the impact of atmospheric deposition fluxes of N, Fe, and P on the marine productivity. PISCES (Pelagic Interactions Scheme for Carbon and Ecosystem Studies v. 2), as a part of the Nucleus for European Modelling of the Ocean (NEMO), includes a detailed representation of the lower trophic levels of marine ecosystems. The model simulates the biogeochemical cycles of carbon and the main nutrients (N, P, Fe, and Si), assuming a constant Redfield ratio (i.e., C:N:P = 122:16:1) in organic matter and living biomass. The external nutrient sources from atmospheric deposition, rivers, sea ice, sediment dissolution, and hydrothermal vents are also considered. PISCES includes two types of phytoplankton, namely calcareous nanophytoplankton and siliceous diatoms, and it simulates the chlorophyll concentrations and the phytoplankton growth based on nutrients' availability (i.e., DP, DN, and DFe for nanophytoplankton and DP, DN, DFe, and DSi for diatoms), temperature and light. Phytoplankton can be grazed by zooplankton or enters directly into the detritus pool. All particulate organic matter sinking to the bottom undergoes remineralization and the nutrients formerly incorporated during photosynthesis are released again. Thus, PISCES simulates the full inorganic carbon cycle including the biological and the carbonate counter pump. At the ocean surface, air-sea gas exchanges for carbon dioxide, oxygen, and nitrogen are parameterized following Wanninkhof (1992). The model has been successfully tested against the response of oceanic productivity to dust (e.g., Guieu et al., 2014) and climatic variability (e.g., Schneider et al., 2008).

### 2.1 Model set-up

### 2.1.1 Physical Ocean forcing

The dynamical physical outputs used to force PISCES for this study were produced by the physical ocean model NEMO, following the protocol of the OMIP simulation (Ocean Modelling Intercomparison Project; Orr et al., 2017). OMIP aims at harmonizing forcing fields of boundary conditions, as well as validation and analysis procedures among different ocean models. Atmospheric forcing fields are from the CORE II forcing (Coordinated Ocean-ice Reference Experiments - Phase II; Large & Yeager, 2009). CORE II provides a 62-year interannual forcing for the period 1948-2009. The physical model is initialized with gridded observational data from the World Ocean Atlas 2013 (Locarnini et al., 2013; Zweng et al., 2013) and then ran for 310 years by repeating the 62-year CORE II forcing. The necessary physical variables to force the offline biogeochemical model PISCES (see Table S1 in the Sup. Mat.) were taken from the last 62-year iteration. To avoid, however, any long-term trends from the spin-up, the multi-year (1948-2009; i.e., the 5th iteration of the 310-year run) mean of daily forcing fields was calculated. The resulting mean 1-year forcing, thus, contains the mean seasonal cycle and is (repeatedly) applied to drive all simulations with the biogeochemical PISCES offline model. All biogeochemical simulations are initialized and forced with the same physical fields from the average 1-year forcing derived from the OMIP run. Thus, all the PISCES

offline simulations are drift-free in physical variables. More details of the OMIP protocol can be found in Orr et al. (2017) and a first validation of the OMIP run is provided by Skyllas et al. (2019).

## 2.1.2 The ocean biogeochemistry model

For this study, PISCES uses a ~1° horizontal resolution with a latitudinal grid refinement in the tropics and 75 layers for the ocean (i.e., ORCA R1) and a timestep of 2700 sec. For the initialization of the ocean biogeochemical fields, the climatological fields of oxygen, nitrate, silicate, and phosphate from the World Ocean Atlas 2009 (WOA; Garcia et al., 2010a, 2010b) along with dissolved inorganic carbon (DIC) and alkalinity from the Global Ocean Data Analysis Project (GLODAP; Key et al., 2004) are adopted. The default N, Fe and P atmospheric nutrient inputs of PISCES in the EC-Earth configuration have been replaced for this work by state-of-the-art monthly mean atmospheric deposition fields recently published in the literature. Note that the default PISCES configuration (Aumont et al., 2015) uses yearly resolution N deposition fluxes of ~67 Tg-N yr$^{-1}$ into the global ocean (Duce et al., 2008) assuming that all deposited N into the ocean to be dissolved. Respectively, the Fe, P, and Si atmospheric inputs were calculated from the same (monthly resolution) dust deposition field. Specifically, for the Fe atmospheric input, the Fe content in dust was set to 3.5%, and its soluble fraction was derived based on the simulated monthly resolved Fe solubility fields (Luo et al., 2008; Mahowald et al., 2009) overall, resulting in a soluble Fe input to the ocean of ~0.15 Tg-Fe yr$^{-1}$. For the P atmospheric input, the P content in dust was set globally to 750 ppm and with a constant solubility of 10% (Mahowald et al., 2008), resulting in a DP deposition flux of ~0.02 Tg-P yr$^{-1}$ in the global ocean. Finally, for Si atmospheric inputs, a constant fraction of 30.8% in dust was set, assuming that 7.5% of the deposited total Si as soluble (~6.35 Tg-Si yr$^{-1}$) and, thus, upon deposition entered in the dissolved silicate pool of the model.

In contrast to previous studies (e.g., Aumont et al., 2015), the new N, Fe, and P atmospheric deposition fields considered here (see Sect. 2.2) are all calculated based on:

1) emissions of natural and combustion nutrient containing aerosols,
2) detailed atmospheric gas- and aqueous-phase chemical schemes, and
3) mineral dissolution processes due to atmospheric acidity and organic ligand (oxalic acid) in aerosol water and cloud droplets.

Note that, as for the default PISCES configuration, Si deposition fluxes into the ocean are only based on the new dust deposition fields coupled in the model. Moreover, for this work, no extra optimizations for the iron scavenging parameters have been applied, since the default PISCES configuration already considers a variable iron solubility on the dust deposition inputs (Aumont et al., 2015). However, the simple chemistry scheme available in PISCES is here used, which is based on one ligand (L) dissolved inorganic Fe and one dissolved complexed iron (FeL). The ligand concentration in the ocean is kept constant, equal to 0.6 nmol L$^{-1}$ and the scavenging rate by dust is equal to 150 d$^{-1}$ mg$^{-1}$ L (see Aumont et al., 2015 and ref. therein). For clarity, we further note that the ocean and biogeochemistry modules used for this study may slightly differ from the version recently used in EC-Earth CMIP6 simulations since at the time the simulations were carried out the final version of EC-Earth was still not released.

Two transient simulations from 1651 to 2100 are here performed to study the impact of nutrients (N, P, and Fe) atmospheric input on global marine productivity:

1) a standard (STD) simulation accounting for the inorganic fractions of the deposited atmospheric nutrients (N, P, and Fe) into the global ocean, and

2) a sensitivity (ORG) simulation, as for STD but also accounting for the organic fractions of the deposited atmospheric nutrients (N, P, and Fe).

We here present results for the preindustrial (PAST: 1851–1870 average), present-day (PRESENT: 2001–2020 average), and future projected (FUTURE: 2081–2100 average) periods. For all PISCES simulations, the first 200 yrs. (i.e., 1651-1850) are not interpreted but considered as a spin-up to reach a quasi-equilibrium state in the model with a well-ventilated upper ocean. Moreover, the atmospheric $CO_2$ mixing ratio is set to the preindustrial value of 284.7 ppm, to effectively isolate the impact of atmospheric deposition on the marine biogeochemistry parameters. All monthly atmospheric inputs of nutrients are prescribed to the model, with the deposition fluxes for the spin-up fixed at the levels calculated with the anthropogenic and biomass burning emissions of the year 1850. To account, however, for potential drifts in the deeper ocean layers, a control (CTRL) simulation as for STD but using only preindustrial (i.e., the year 1850) atmospheric nutrients (N, P, and Fe) inputs into the global ocean is also performed. Figure S1 (see Sup. Mat.) demonstrates that for the main ocean basins the drift in vertically integrated primary production is minimal and clearly below the signal imposed by the altered nutrient deposition after 1850. This holds even for the Southern Ocean where the impact of atmospheric deposition is typically weak due to the absence of neighboring emission sources. Nevertheless, all model results presented in this work have been adjusted by subtracting the drift of the control run from STD.

## 2.2 The atmospheric nutrient inputs

All atmospheric nutrient inputs coupled to PISCES are derived from the offline global atmospheric Chemistry-Transport Model (CTM) TM4-ECPL. The CTM is driven by the ECMWF (European Centre for Medium-Range Weather Forecasts) Interim reanalysis project (ERA-Interim) meteorology (Dee et al., 2011) for the year 2010, and uses a horizontal resolution of $3^o$ in longitude by $2^o$ in latitude, with 34 hybrid layers up to 0.1 hPa. The CTM simulates the gas-phase chemistry along with the major non-methane volatile organic compounds, as well as all major aerosol components, such as dust, sea-salt, organic aerosols (OA), black carbon (BC), $SO_4^{2-}$, $NH_4^+$, and $NO_3^-$. The thermodynamic equilibrium model ISORROPIA II (Fountoukis and Nenes, 2007) is used to estimate the water content and the acidity of hygroscopic aerosols, accounting also for the impact of crustal materials (i.e., $Ca^{2+}$, $Mg^{2+}$, $K^+$, $Na^+$, $Cl^-$) from mineral dust and sea-salt (Myriokefalitakis et al., 2015). The in-cloud acidity is mainly controlled in the model by strong acids, i.e., sulphuric acid $H_2SO_4/SO_4^{2-}$, methanesulphonic acid (MSA/MS$^-$), and $HNO_3/NO_3^-$, and bases ($NH_3/NH_4^+$), accounting also for the dissociation of hydrated $CO_2$, sulfur dioxide ($SO_2$), and oxalic acid (Myriokefalitakis et al. 2011). The CTM further considers the multiphase chemistry secondary aerosol production in cloud droplets and aerosol water (Myriokefalitakis et al., 2011), as well as the secondary organic aerosol (SOA) formation via gas-to-particle partition over land and oceanic regions (e.g., Myriokefalitakis et al., 2010). The anthropogenic (including

ships and aircraft emissions) and biomass burning emissions from the historical Atmospheric Chemistry and Climate Model Intercomparison Project (ACCMIP) database (Lamarque et al., 2013) are used in the CTM, and the Representative Concentration Pathways 8.5 (RCP8.5) scenario (van Vuuren et al., 2011) is applied for the future CTM emissions. For all CTM simulations, a spin-up period of one year is applied. Note that we use here nutrient atmospheric deposition fields based on simulations with the latest updates of the CTM, as recently published by Kanakidou et al. (2020), which are all based on an improved on-line dust emission scheme (Tegen et al., 2002). As a result, the global dust source for all simulations is here equal to $\sim$1287 Tg yr$^{-1}$ for the year 2010, well comparable to a multimodel estimate of $\sim$1257 Tg yr$^{-1}$ as reported by Huneeus et al. (2011). For example, the global annual mean Si deposition input rate to the global ocean, which is solely related to the dust deposition, is accounted here to about 4.7 Tg-Si yr$^{-1}$ and in the range of other estimations when a solubility of 7.5% is considered (e.g., see Krishnamurthy et al., 2010). Nevertheless, some small differences on the total amount of deposited nutrients over the ocean compared to previously published results (e.g., Kanakidou et al., 2016; Myriokefalitakis et al., 2015, 2016) are expected, due to the various updates of the code, the different year of simulation, or even the different definitions of the oceanic regions due to the applied horizontal analysis in PISCES (i.e., the ORCA R1).

Simulations with the atmospheric transport and chemistry model are, nevertheless, extremely expensive. Therefore, limitations in available computational resources made it necessary to reduce the CTM simulations to representative single years for 1) the preindustrial state (before 1850), 2) the present-day state (representing the year 2010), and 3) a mid-century (2050), as well as, an end of century (2100) state. However, as the typical residence time of tropospheric aerosols is in the order of a days, the atmospheric depositional fields used in PISCES represent a well equilibrated atmospheric chemistry and deposition flux, without the need of time transient simulations.

For the ocean biogeochemistry model spin up (i.e., from 1651 to 1850) the preindustrial field (the year 1850) was applied. After the 200 years spin-up period, the atmospheric deposition input data for the STD and ORG simulations were linearly interpolated from preindustrial to present-day conditions (i.e., the year 2010) to smoothly capture the transition from past to the modern conditions (e.g., Krishnamurthy et al., 2009). Respectively, the deposition data from the present day were linearly interpolated to the projected estimates (i.e., the years 2050 and 2100). Note that for all temporal and spatial interpolations of this work (as well as, for the drift corrections), the Climate Data Operators (CDO v.1.9.8) software, as provided by the Max Planck Institute for Meteorology, is here used (https://code.mpimet.mpg.de/projects/cdo/embedded/cdo.pdf; last access 29/02/2020). An example of the globally averaged N, Fe, and P atmospheric deposition data as simulated by the CTM and applied in PISCES is presented in Fig. S2. Overall, the here discussed simulations should be considered as idealized sensitivity experiments to estimate the response on the ocean surface biogeochemical properties to changed atmospheric deposition.

**2.2.1 Nitrogen**

For the calculation of the atmospheric nitrogen deposition fluxes, the CTM uses primary emissions of NOx, NH$_3$, marine amines, and emissions of particulate organic nitrogen (ON) from natural and anthropogenic sources (Kanakidou et al., 2012, 2016, 2018). The particulate ON is linked to the OA tracers using varying N:C molar ratios (Kanakidou et al., 2012), as well

as to the SOA formation under high NOx-to-VOC conditions (Myriokefalitakis et al., 2010; Tsigaridis et al., 2006). Amines of marine origin in the gas phase are also considered to form amine salts (Myriokefalitakis et al., 2010). A more detailed description of the N-cycle parameterization in the CTM can be found in Kanakidou et al. (2016, 2018).

Figure 1b presents the annual mean spatial distribution of dissolved nitrogen deposition as considered in PISCES for the STD simulation. The present-day DIN deposition fluxes into the global ocean are estimated to be ~40 Tg-N yr$^{-1}$ (Table 1). DIN deposition (oxidized and reduced inorganic nitrogen) shows the highest fluxes downwind industrial areas of the Northern Hemisphere and the tropical biomass burning regions due to the enhanced NOx emissions, as well as downwind Europe, China, and Indonesia, reflecting overall the high strength of NH$_3$ emissions in these regions. DIN deposition exceeds $1 \cdot 10^{-3}$ kg-N m$^{-2}$ yr$^{-1}$ downwind the eastern United States, Europe, India, China, and Indonesia (Fig. 1b). Some low DIN deposition fluxes over the remote ocean are related to the recycling of the marine NH$_3$ sources. Moreover, when the ON is accounted for the dissolved nitrogen deposition, as considered in the ORG simulation (Fig. S3a), much higher nitrogen deposition fluxes are calculated in the tropics mainly due to the large contribution by primary biogenic particles, the biomass burning emissions, along with the ON production during SOA formation (Fig. S3b). For comparison, we note that the total river supply of bioaccessible nitrogen in the model is roughly 36 Tg-N yr$^{-1}$ (Aumont et al., 2015). Note, however, that the total present-day dissolved nitrogen deposition estimate (~58 Tg-N yr$^{-1}$ for the ORG simulation; Table 1) is lower compared to the N deposition fluxes (~67 Tg-N yr$^{-1}$) as taken from Duce et al. (2008) and used in previous PISCES configurations (Aumont et al., 2015).

Compared to the present day, almost all ocean basins (except some parts of the South Indian and the South Pacific Ocean) display a substantially lower (>50%) nitrogen deposition flux during the preindustrial era (Fig. 1a). Inorganic nitrogen inputs to the ocean have significantly increased for present-day along the coasts of the African, Australian, and the South American continents, downwind densely populated areas in the northern hemisphere, such as the east coast of North America and Europe, as well as regions with intensive agriculture downwind the coast of East and South Asia (Kanakidou et al., 2016). Globally, present-day atmospheric inorganic nitrogen inputs to the ocean have increased by a factor of ~2 since 1850 (Table 1) due to the respective increase of NH$_3$ and NO$_x$ emissions. The projected atmospheric DIN inputs to the ocean indicate also a decrease (~15%), although much less significant compared to preindustrial times (Table 1) since the reduction in NOx emissions is projected to be compensated by the continuing increase in NH$_3$ emissions. Note that the preindustrial ON deposition fluxes into the ocean are estimated to be roughly of the same magnitude as the inorganic nitrogen oceanic input (i.e., ~15 vs. ~20 Tg-N yr$^{-1}$; Table 1). Projections under the RCP8.5 scenario, however, indicate that NHx emissions will gain in importance, resulting overall in a weaker contribution of the oxygenated inorganic nitrogen to the atmospheric deposition into the ocean for future conditions.

**2.2.2 Iron**

The global atmospheric Fe deposition fluxes are parameterized in the CTM considering primary Fe emissions associated with minerals in dust and combustion processes. The Fe content of dust minerals is based on detailed mineralogy maps (Nickovic et al., 2012), accounting also for an initial soluble Fe content in the mineral emissions (Ito and Xu, 2014), overall resulting in

a mean Fe content of about 3.2% in dust emissions. For the Fe-containing combustion aerosols, the CTM accounts for emissions from biomass burning, coal, and oil combustion (Ito, 2013; Luo et al., 2008) with dissolved Fe contents of 12% for biomass-burning, 8% for coal combustion, and 81% for oil combustion from shipping Fe emissions. The CTM further accounts for acid- and organic ligand-solubilizations of dust aerosols, in both aerosol water and cloud droplets, as well as, for the aging (i.e., the conversion of insoluble to soluble) of the Fe-containing combustion aerosols via atmospheric processing. More details on the atmospheric Fe-cycle set-up can be found in Myriokefalitakis et al. ( 2015, 2018), along with updates from Kanakidou et al. (2020).

Figure 1e presents the annual mean spatial distribution of dissolved iron deposition fluxes, as considered from PISCES for the STD simulation. DFe deposition fluxes into the ocean present strong spatial variability (Fig. 1e) with an annual flux of ~0.28 Tg-Fe yr$^{-1}$ for the STD and ~0.35 Tg-Fe yr$^{-1}$ for the ORG simulation (Table 1), both estimates are well in the range of the mean DFe deposition flux into the ocean (0.2-0.4 Tg-Fe yr$^{-1}$), as derived from the GESAMP model intercomparison study (Myriokefalitakis et al., 2018). For comparison, we note that the total riverine Fe supply in the model equals 1.45 Tg-Fe yr$^{-1}$ (see Aumont et al., 2015 and ref. therein). The highest annual mean DFe deposition fluxes into the ocean (Fig. 3c) occur downwind dust source regions with significant deposition rates to also be found in the outflow of tropical biomass burning regions (i.e., Central Africa and Indonesia), reflecting the importance of combustion processes. Annual mean DFe deposition rates of ~$10^{-5}$ kg-Fe m$^{-2}$ yr$^{-1}$ are considered for the tropical Atlantic Ocean, as well as, for the Indian Ocean under the influence of the Arabian and Indian peninsulas. For the Southern Ocean, the DFe atmospheric inputs (up to ~$10^{-6}$ kg-Fe m$^{-2}$ yr$^{-1}$) are mainly associated with the Patagonian, the Southern African, and Australian deserts. Figure S3c further presents the annual mean DFe deposition fluxes into the ocean when the organic fraction is considered. The organic bound Fe is produced in the CTM during the organic-ligand dust dissolution processes (i.e., as Fe(II/III)-oxalato complexes), and further accounted for a fixed 0.1% fraction in Fe-containing combustion aerosols (Kanakidou et al., 2018; Myriokefalitakis et al., 2018). Note that although this estimate is highly uncertain due to the lack of observational data, and it overall appears to contribute modestly to the global DFe atmospheric input to the ocean (~0.07 Tg-Fe yr$^{-1}$), the deposition of organic bound Fe aerosols can be potentially important (~up to 40%) in the remote tropical Pacific and the Southern Ocean (Fig. S3d) where the atmospheric Fe concentrations are extremely low and are mainly occurring due to biomass burning and anthropogenic (i.e., ship) combustion emissions (Ito et al., 2019a).

The CTM calculates increases in DFe deposition rates since preindustrial times, as stronger Fe combustion emissions and a more efficient dust dissolution rate due to the more acidic modern environment occur in the modern era; overall, accounting for about 1.5 times higher DFe atmospheric input to the global ocean for present-day (Table 1). On the other hand, the derived DFe in the ocean for the future atmosphere is calculated ~14% lower than present-day conditions (Table 1), owing to the projected changes in anthropogenic emissions and air-quality. For the preindustrial era, the largest differences in atmospheric DFe deposition fluxes compared to present-day are considered in the Northern Indian, the North Pacific, and the tropical North Atlantic Oceans (Fig. 1d). Lower PAST atmospheric DFe deposition fluxes compared to present-day are simulated in remote oceanic regions away from dust plumes, in the tropical and subtropical Pacific Ocean, also due to the increased present-day

atmospheric processing. The combustion source of DFe turns out, however, to be rather important near industrial and biomass burning sources, such as downwind South and East Asia, where dust emissions are lower. For the future emissions, however, smaller changes are derived, with a general decrease of the atmospheric DFe input in most parts of the global ocean (Fig. 1f), except for some increases in the Eastern North Pacific Ocean and over the very low atmospheric Fe concentrations regions in

the remote Southern Ocean. Nevertheless, the demonstrated increases in the past (and future) deposition fluxes over remote oceanic regions with low atmospheric Fe concentrations, such as the Southern Ocean or downwind strong biomass burning and anthropogenic coal combustion regions, are due to both the changes in emission strengths and the impact of atmospheric processing on the dissolved Fe lifetime (Myriokefalitakis et al., 2018).

### 2.2.3 Phosphorus

The atmospheric P-cycle is calculated based on emissions of insoluble mineral-P, phosphate, and insoluble and soluble organic P (OP), with the resulted DP deposition fluxes in the CTM being driven by natural (i.e., dust, bioaerosols, sea spray and, volcanic aerosols) and combustion P-containing aerosol emissions. Acid solubilization of dust particles (i.e., conversion from mineral P to phosphate) takes place in both aerosol water and cloud droplets, along with the conversion from insoluble to soluble OP aerosols during atmospheric processing. The CTM accounts for two P-containing insoluble minerals (the

fluoroapatite and the hydroxyapatite) in dust based on soil mineralogy maps (Nickovic et al., 2012), as well as for OP present in soil's organic matter (Kanakidou et al., 2012). A solubility of 10% is applied to all P-containing dust emissions in the CTM. For P-containing combustion aerosols, the CTM accounts for anthropogenic (i.e., for fossil fuel, coal, waste, and biofuel) and biomass burning emissions, based on observed P/BC mass ratios (Mahowald et al., 2008). Sea-spray and volcanic aerosols account for a rather low DP global source, in contrast to bioaerosols which are estimated to contribute significantly to DOP.

The naturally emitted OP by bioaerosols is overall represented by bacteria, fungi, and pollen (Myriokefalitakis et al., 2017). More details on the P-cycle representation in the CTM can be found in Myriokefalitakis et al. (2016).

The present-day global annual DIP deposition flux into the global ocean for the STD simulation accounts for ~0.10 Tg-P yr$^{-1}$ (Table 1), presenting also a strong spatial variability (Fig. 1h). The highest DIP deposition input rates occur downwind the major dust source regions (~$5 \cdot 10^{-6}$ kg-P m$^{-2}$ yr$^{-1}$) owing to the phosphate content in dust emissions. High DIP deposition

fluxes (~$1 \cdot 10^{-6}$ kg-P m$^{-2}$ yr$^{-1}$) also occur downwind heavily forested regions (i.e., Amazonia, Central Africa, and Indonesia) due to the enhanced biomass burning sources. The combustion of anthropogenic origin further contributes to the DIP deposition flux into the ocean, such as downwind the South and East Asia (~$1 \cdot 10^{-6}$ kg-P m$^{-2}$ yr$^{-1}$). Notable deposition rates are also illustrated away from dust sources, such as in the Northern Atlantic and Pacific Ocean (~$5 \cdot 10^{-7}$ kg-P m$^{-2}$ yr$^{-1}$), due to the mineral P solubilization during atmospheric transport and somewhat lower rates occur in the Southern Ocean. Moreover, the

consideration of organic fraction to atmospheric DP inputs to the global ocean (Fig. S3e) results in a ~50% increase in the global DP deposition flux (Table 1), but stronger regional increases of up to 80% can be also seen (Fig. S3f). Note also that in PISCES, DP fluxes of roughly 3.7 Tg-P yr$^{-1}$ is also delivered to the ocean by rivers (Aumont et al., 2015).

An increase in the global DIP deposition of ~40% is considered here for present-day (i.e., 0.06 Tg-P yr$^{-1}$) compared to preindustrial conditions, and a modest decrease of ∼20% is projected for the year 2100 (i.e., 0.08 Tg-P yr$^{-1}$ under the RCP8.5 scenario), as shown in Table 1. The regional differences for the preindustrial times appear, however, to be stronger (up to 60 %; Fig. 1g), especially downwind highly populated regions of the Northern Hemisphere, in the Atlantic and Pacific Oceans. DIP deposition fluxes are also projected to decrease over the midlatitudes of the Northern Hemisphere where human activities dominate under the RCP8.5 scenario, with the largest changes up to 40% downwind of China and Australia. Significant changes (~20 %) are also illustrated in the Arabian Sea and the Bay of Bengal due to the expected increases in population (Fig. 1i).

## 3 Results

### 3.1 Oceanic nutrient concentrations

**Nitrate**: The simulated annual mean nitrate surface concentrations in the seawater for the present-day and the relative differences for past and future eras are presented in Fig. 2. The present-day surface nitrate distribution shows high concentrations along the equatorial divergence where nutrients are upwelled, and solar insolation supports good light conditions throughout the year. In the high latitudes, cooler water temperatures and seasonally damped light conditions reduce nutrient consumption by biological productivity, resulting overall in elevated annual mean nutrient concentrations. High surface concentrations are also calculated in the high latitude Southern Ocean where the deep convection around Antarctica maintains high nutrient transports to the surface and productivity is limited by a thick mixed layer, lower water temperatures, and reduced light conditions. Elevated nutrient concentrations are likewise simulated in the Eastern Equatorial Pacific, and the Subarctic North Pacific, i.e., the well-known HNLC regions. All in all, this reflects a reasonable simulation of the abiotic oceanographic drivers in the model. A comparison between the simulated present-day surface nitrate concentration and the compiled data from the World Ocean Atlas is given in Fig. S4.

Increased atmospheric nitrogen deposition fluxes from the preindustrial to the modern era (Table 1) result in a respective increase in surface nitrate concentrations in almost all oceanic regions, with some exceptions in the Eastern Equatorial and the subpolar Pacific Ocean (Fig. 2a). In remote oceanic areas, far from any coastal or riverine nutrient supply and thus strongly nutrient-limited, the higher present-day inorganic nitrogen and iron atmospheric inputs compared to the preindustrial era (Figs. 1a,d) increase primary production (see also Sect. 3.3). This may, overall, lead to increased export of the surface seawater nitrogen to the deeper ocean in the form of sinking biogenic particles in these oligotrophic oceanic regions (e.g., Krishnamurthy et al., 2007, 2010). For the future conditions, however, the model calculates both negatives and positive changes in the surface nitrate concentrations, resulting from an overall decrease (~4%) of the global inorganic nitrogen oceanic input (from atmospheric deposition and N$_2$-fixation) (Table 1). Indeed, for the future era, a decrease in almost all oceanic regions is calculated, except for the coastlines of Southeast Asia, Africa, and South America (Fig. 1c).

**Iron**: Figure 2 also presents the annual mean surface concentrations of iron for present-day in STD simulation along with the respective past and future relative differences. Present-day surface iron distribution (Fig. 2e) shows high concentrations in the subtropical North Atlantic Ocean and the Arabian Sea, with the lowest surface concentrations being calculated in the Eastern Equatorial Pacific and the Southern Ocean. A secondary maximum in iron concentrations is calculated in the Subarctic North Pacific. In general, iron concentrations in the model are low, especially in the Southern Ocean, the Eastern Equatorial Pacific, and the Subarctic North Pacific. Higher concentrations are found along the coasts or over the continental margins. For the past (Fig. 2d) and future (Fig. 2f) conditions, the model generally calculates lower iron surface concentrations, reflecting overall the respective decreases in DFe deposition fluxes into the ocean (Figs. 1d,c, respectively). Consequently, the strongest declines are found in the northern hemisphere, especially in the mid to high latitude Pacific and Atlantic. An exception is the NW Pacific where higher iron input in FUTURE (Fig. 1f) results in elevated oceanic iron concentrations. A comparison of the simulated present-day surface oceanic iron concentration with available DFe oceanic observation data (Tagliabue et al., 2012) is given in Fig. S4.

**Phosphate**: The annual mean present-day phosphate surface concentrations as calculated for the STD simulation together with the respective relative differences for past and future eras are presented in Figs. 2g-i. In general, the phosphate concentrations show similar distributions as the dissolved inorganic nitrogen. The surface phosphate distribution in the model shows high concentrations along the equatorial divergence where nutrient-rich deep water is upwelled, as well as in the high latitudes, with the highest surface concentrations to be simulated in the Southern Ocean. A secondary maximum is calculated in the Eastern Equatorial Pacific and the Subarctic North Pacific; both regions are subject to large scale oceanic upwelling of deep waters. Note that in general, phosphate concentrations in deep waters are higher than at the surface where nutrients are removed by biotic productivity and exported by sinking particulate organic matter to the deeper ocean. However, due to the constant Redfield Ratio (i.e., C:N:P = 122:16:1) applied in the model (Aumont et al., 2015), the phosphorus cycle is closely related to that of nitrogen, as both are subjected to the same large-scale physical processes and circulation in the ocean. An exception is the $N_2$-fixation that acts as an additional external source for inorganic nitrate in the model. A comparison of the simulated surface phosphate concentration with observational data is given in Fig. S6.

Despite the roughly 1.7 times increase in the phosphate deposition inputs to the ocean from preindustrial to the modern era (Table 1), the preindustrial surface phosphate oceanic concentrations are calculated 20-50% higher in most oceanic regions (Fig. 2g), except for the Southern Ocean where no significant change is calculated. Accordingly, although the projected decrease of the global phosphate input (Table 1), higher phosphorus surface oceanic concentrations are simulated for the future, up to ~20% (Fig. 2i). The main reason for these elevated phosphate concentrations is the decreased primary production almost everywhere (Figs. 3d,f). Indeed, as nitrogen is the limiting factor for phytoplankton in the open ocean, the primary production rates have been lowered according to lower nitrogen deposition rates. Accordingly, less phosphate consumption by phytoplankton growth takes place and this outcompetes the effect of lowered P deposition, leading to relatively higher phosphate concentrations. The effect of the decreased productivity on phosphate concentrations would be, however, even stronger if it was not being partly compensated by higher $N_2$-fixation rates. Overall, this points to the marine biogeochemical

processes as an essential factor in controlling the phosphorus concentrations at the surface ocean rather than depositional fluxes (see also Sect. 3.3.1).

## 3.2 Nitrogen fixation

For the STD simulation, the nitrogen fixation is calculated to about 112 Tg-N yr$^{-1}$ for the present day (Table 1). The respective relative differences compared to past and future periods are also presented in Figs. 3a-c. Compared to modern times, the model calculates a significant decrease (up to 20%) in nitrogen fixation in the tropical and subtropical Pacific and the subtropical Atlantic Ocean for the preindustrial era. On the contrary, downwind land sources such as the Bay of Bengal and Indonesia, nitrogen fixation is higher for the preindustrial conditions, due to the lower nitrogen deposition fluxes accounted by the model (see Sect. 2.2.1). Finally, the nitrogen fixation rates present very low differences in the Equatorial Pacific, Equatorial Atlantic, and the Southern Indian Ocean (0-10%) away from land sources (Fig. 3a). Note, however, that nitrogen fixation in PISCES is restricted to warm waters (i.e., above 20$^{o}$ C). Therefore, the strong reductions of nitrogen deposition in the mid to high latitude North Pacific in PAST have no direct impact on nitrogen fixation rates. In the subtropical Pacific reduced nitrogen fixation rates mainly reflect the diminished iron input (Fig. 1d). On a global scale, the model calculates overall only a small decrease (~0.2%; Table 1) in preindustrial nitrogen fixation rates compared to present-day, mainly as a result of the decreased soluble iron inputs in the subtropical North Pacific (Fig. 1d). For the future conditions, the model likewise calculates a modest decrease in the global nitrogen fixation (~1%; Table 1) along with decreased iron inputs to the ocean (Figs. 1c,f, respectively) resulting overall in some lower rates of up to 10% in the Equatorial Pacific Ocean (Fig. 3c).

## 3.3 Primary production

The present-day annual mean primary production together with the relative differences to past and future periods are presented in Figs. 3d-f. The primary production distribution in the open ocean shows high rates along the equatorial divergence and in the high latitudes, where nutrients concentrations are high. The decreased nitrogen deposition during preindustrial conditions compared to present-day results in lowered primary production rates almost in all oceanic regions. A projected modest decrease of primary production rates is calculated by the model (Fig. 3f), owing also to the lower (~14%) dissolved iron deposition fluxes implemented from present to future conditions (Fig. 1f).

The present-day modeled globally integrated production (~47 Pg-C yr$^{-1}$; Table 1) is lower compared to satellite-based estimates from SeaWiFS (Behrenfeld et al., 2005), obtained equal to ~60-67 Pg-C yr$^{-1}$, but in the range of estimates from other studies (e.g., 23.9 - 49.1 Pg-C yr$^{-1}$; Steinacher et al., 2010). A more detailed comparison between satellite estimates (Behrenfeld et al., 2005) and model simulations of the global primary production is presented in Fig. S7. The simulated primary production reproduces the main features derived from satellite-based observations (Figs. S7a,d). The model, however, simulates relatively low rates (~200 mg-C m$^{-2}$ day$^{-1}$) in the subtropical gyres and higher rates (> 500 mg-C m$^{-2}$ day$^{-1}$) in upwelling regions, the North Atlantic and the Southern Ocean (Fig. S7g). Nevertheless, as also known from other modeling studies, the primary production in the tropics might be overestimated, whereas in the higher latitudes it might be underestimated (e.g., Steinacher

et al., 2010). High productivity in the open ocean areas is linked to upwelling areas such as the equatorial divergence zones or coastal upwellings, like at the west coasts of South Africa or northern South America. Primary productivity rates are also underestimated in the North Atlantic region (Fig. S7g), probably due to the low accumulation of nutrients during winter as already discussed. Figure S7 further illustrates the simulated primary production for the boreal winter (DJF) and summer (JJA),

compared to the respective observation-based data. The model generally compares reasonably for both seasons; however, the modeled primary production rates (Figs. S7e,f, respectively) are calculated lower in the high latitudes. The pronounced seasonality due to light limitation and temperature of phytoplankton growth in the North Atlantic is well captured by the model. During the warm season, higher water temperatures and better light conditions increase primary production and nutrient consumption by phytoplankton growth in the high latitudes. This results in depleted nutrient concentrations in the North

Atlantic during boreal summer. In the North Pacific, which is known as an HNLC region, nutrients remain at higher levels due to insufficient iron support. This is somewhat underestimated in the model as the simulated nitrate concentrations are too low (Fig. S4f), probably related to the slightly high iron concentrations in the upper 100 m North Pacific (Figs. S5d,f). The same biases mentioned for phosphate are likewise seen in the nitrate concentration as most features are mainly controlled by the model physics. Overall, only nitrogen fixation and denitrification can modulate the nitrate concentrations, apart from all other

processes that also influence the phosphate in the same way (e.g., productivity, remineralization dissolution, etc.).

### 3.3.1 Role of depositional nutrient elemental ratios

Despite the relatively strong changes in total atmospheric nutrient supply from PAST to FUTURE (Table 1), the impact of atmospheric nutrients on the global productivity rates remains low in the model. This is, nevertheless, not unexpected, as the atmospheric nutrient supply constitutes only a small fraction of the total ocean nutrient inventory. In addition, oceanic regions

that are not nutrient-limited today are less sensitive to external nutrient supply. Finally, a large part of primary production is regenerated by remineralized nutrients from particulate organic matter (mainly detritus) in the upper ocean layer.

To further identify the oceanic regions that are particularly sensitive to changes in external nutrient inputs, the limiting factors for local productivity in the model are investigated. Figure 4a displays limitations due to nitrogen or phosphorus. High values, indicating low limitation, are seen in regions that are subject to intense upwelling, like in the equatorial divergence zones or

the western margins of NW and southern Africa and South America (coastal upwelling). Accordingly, these regions are less sensitive to atmospheric deposition as nutrients are supplied from deep ocean layers. Lower nutrient limitation is likewise seen in the mid to high latitudes where limitations by temperature and light (Fig. 4b) limit the growth rates. Exceptions are the North Pacific, the Southern Ocean, and the equatorial Pacific where iron limitation matters (Fig. 4c). Consequently, the model's nutrient sensitivity is largest in the subtropics, in particular in the subtropical gyres where good light conditions and warm

waters support high growth rates, paralleled by diminished nutrient supply from depth due to Ekman pumping. Furthermore, these regions are far from land nutrient sources and so, a major part of total primary production relates to regenerated production (i.e., with low rates of external nutrient supply) which is limited by nutrients as temperatures and light are sufficient. This makes productivity in the subtropical gyres sensitive to changes in the external atmospheric nutrient.

In regions with significant macronutrient limitations, the elemental ratio of deposited N:P can be, however, important. To estimate the relative impact of the changes in this ratio, we calculated the modeled nitrogen concentrations relative to the model's Redfield ratio (Fig. 5a). For PRESENT, the model exhibits almost everywhere a deficiency with respect to nitrogen (except for some coastal areas). This is in good agreement with data from WOA which likewise indicates a predominant nitrogen deficiency almost everywhere (Fig. 5b). Next, the N:P ratio relative to the Redfield as supplied by atmospheric deposition for PRESENT (middle) together with the changes in PAST and FUTURE (Fig. S8b) is derived. Overall, a strong excess of N compared to P for modern times is indicated. As a consequence of the model's nitrogen deficiency (Fig. 5a), however, this atmospheric nitrogen excess maintains higher productivity than without the atmospheric supply. For preindustrial times, the atmospheric N:P ratio is almost everywhere reduced, increasing thus the N-deficiency. Hence, rather the lowered atmospheric nitrogen inputs than the lowered phosphorus inputs in PAST and FUTURE are responsible for the diminished productivity in these experiments. To further demonstrate this, we carried out an additional sensitivity simulation (namely PIP simulation) where the phosphorus atmospheric deposition fluxes kept constant at preindustrial levels, while the other studied atmospheric inputs (i.e., N and Fe) varied as for the STD simulation. As expected, the effect on phosphate concentrations (Fig. S9b) and productivity (Fig. S9d) in this sensitivity simulation remains extraordinarily low, i.e., the relative difference to STD is almost everywhere below 1%. This overall demonstrates that the changes in phosphorus do not play a significant role in marine productivity from preindustrial to future periods.

### 3.3.2 Global patterns of productivity change for preindustrial atmospheric deposition inputs

Model calculations demonstrate three major oceanic regions where the reductions in productivity are strongest in PAST, i.e. the subtropical gyres of the northern hemisphere Pacific, the Atlantic Ocean, and the northernmost North Pacific (Fig. 3d). The subtropical gyres, however, are the most sensitive to changes in nitrogen input and clearly show the strongest productivity reduction compared to PRESENT. Indeed, the nitrogen concentrations in the subtropics are not much affected (Fig. 2a). This is because good light conditions and warm waters persistently maintain high rates of nutrient consumption, so nitrogen concentrations are already very low in PRESENT. Thus, a change in external nutrient supply feeds immediately into productivity without a significant imprint on nitrogen concentrations.

In the northernmost Pacific, the strong productivity decline in PAST (Fig. 3d) is primarily related to the lowered availability of iron, although the reductions in iron deposition remain below 20% (Fig. 2d). However, besides light conditions, iron availability is the most important factor for limiting productivity in this region (Fig. 4c) compared to nitrogen and phosphorus (Fig. 4a). As a consequence, slight changes in iron supply will have a strong impact.

For most of the world ocean, productivity changes in FUTURE are in qualitative agreement with PAST, but less pronounced (Figs. 3d,f). This is mostly because both FUTURE and PAST experiments reflect reduced anthropogenic emissions and thus the same mechanisms are involved. The only exception is the western North Pacific where productivity is rather elevated in FUTURE (Fig. 3f) whereas it is strongly reduced in PAST (Fig. 3d). The different response is related to different iron inputs to the North Pacific (Figs 1d,f).

### 3.3.3 Changes in phytoplankton composition

As already mentioned in Sect. 2, PISCES models two phytoplankton functional types, i.e., the nanophytoplankton producing calcareous shells and the diatoms producing siliceous shells (Aumont et al., 2015). In the high latitudes, a large part of productivity is related to siliceous diatoms (e.g., Malviya et al., 2016; Uitz et al., 2010) which is accounted for in the model by low nanophytoplankton to diatoms ratios (Fig. 6b). Accordingly, the overwhelming part of productivity reduction in the northernmost Pacific (Fig. 3d) is related to the decline of diatoms. This is well reflected by the increase of the nanophytoplankton to diatoms ratio for PAST relative to PRESENT (Fig. 6a). In turn, this leads to enhanced silicate concentrations in the North Pacific (Fig. 7a). Note that the Si atmospheric inputs are solely related to the dust deposition fluxes, and thus they do not have any interannual variability in the model. Part of the unutilized silicate is advected southward via the North Pacific Current and the California Current, leading also to elevated concentrations along the western coast of North America (Fig. 7a). A further consequence of the strongly diminished productivity is an accumulation of nitrogen in the subpolar gyre of the North Pacific (Fig. 2a). The nitrogen anomaly is strongest in the southwestern area of the gyre and part of the excess nitrogen is injected into the northern California Current. As a result, a strong positive and wedge-shaped productivity anomaly develops in front of western Canada in PAST (Fig. 3d). The positive anomaly is caused by the increased production of nanophytoplankton productivity (not shown) which dominates in this region, as indicated by higher nanophytoplankton to diatom ratios (Fig. 6b); i.e., north of the wedge lowered iron limits productivity, while south of the wedge nitrate is limiting. Altogether, this reflects a slight shift from diatom to nanophytoplankton production in the eastern Pacific (north of 40 °N), as indicated by a decline of ~10% of the nanophytoplankton to diatoms concentrations in the upper 20 m (Fig. 7b).

Apart from the northernmost Pacific, the decline in diatom production leads almost everywhere to slightly increased silicate concentrations in PAST (Fig. 7a). Productivity changes in the Southern Ocean remain low (Fig. 3d) for PAST. The reason for this is the strong light limitation around Antarctica (Fig. 4b) and the deep mixed layer which suppresses productivity and subsequently builds up a large pool of unutilized nutrients. Part of the unutilized nutrients is advected further north into the Southern Ocean, driving productivity there. Accordingly, the reduced deposition of nitrogen and iron in this area (Figs. 1a,d) have only a slight impact on productivity. Consequently, this region is relatively robust against external nutrient input maintaining stable productivity. A similar effect is seen for the North Atlantic where vigorous exchange with Arctic waters takes place across the Norwegian-Greenland Sea. By contrast, in the subpolar North Pacific, the import of unutilized nutrients from the Arctic is hampered, as the water exchange with polar waters is limited by the shallow Bering Strait and the Aleutian Arc. Therefore, the North Pacific appears the most sensitive to external nutrient inputs compared to other oceanic regions.

### 3.3.4 Global patterns of productivity change for future atmospheric deposition inputs according to the RCP8.5 emission scenario

For most parts of the global ocean, the changes in productivity in the FUTURE experiment are in qualitative agreement with PAST, but less pronounced. This is mostly because both FUTURE and PAST experiments reflect reduced anthropogenic emission and so the same mechanisms are involved. The only notable exception compared to PAST (Fig. 3d), is demonstrated

in the eastern North Pacific where a strong negative wedge-shaped anomaly is seen in FUTURE (Fig. 3f). This opposite response is related to different iron inputs to the North Pacific. In FUTURE, the reduction of iron atmospheric inputs (Fig. 1f) is by far less strong and in the NE Pacific is even higher than today, thus the productivity increases in the NE Pacific subtropical gyre (Fig. 3f). As a result, more nitrogen is consumed in the subpolar gyre in FUTURE and no nitrogen accumulation takes place as in PAST (Figs. 2a,c). Accordingly, a strong negative nitrogen anomaly develops in the western North Pacific and nitrogen depleted waters are advected southward along the California Current (i.e., opposite to PAST). Altogether, these results imply an extreme sensitivity of the North Pacific against changes in atmospheric iron input. By contrast, the North Atlantic, which is less affected by iron limitation, reflects a widespread decline in productivity mainly controlled by reduced nitrogen inputs.

## 4 Biogeochemistry responses to atmospheric organic nutrients

Most marine biogeochemistry studies mainly account for the inorganic fraction as the most important pool of nutrients from the atmospheric pathway. However, state-of-the-art atmospheric chemistry models nowadays not only efficiently calculate the total dissolved nutrient atmospheric deposition fluxes, but they include the organic part as well, which turns out to be rather important for the total magnitude of the atmospheric input to the ocean (Fig. S3). Great uncertainty still exists concerning the importance of organic fraction on oceanic productivity. For this, we separated here the inorganic and organic fractions of the atmospheric nutrients to investigate the role of organic components in marine biogeochemistry. To estimate the response of the marine ecosystem to the contribution of the organic fraction in the atmospheric nutrient (N, P, and Fe) inputs, the differences in nitrogen fixation and primary production rates between the STD and the ORG simulations are presented in Fig. 8. Note that as for the riverine organic fractions in the model (see Aumont et al., 2015), we assume here an instant transformation of the atmospheric dissolved organic nitrogen (DON) and organic phosphorus (DOP) inputs to the respective inorganic fractions in the water column.

When the organic fraction of the atmospheric nutrients is considered in the model, a modest decrease in the global nitrogen fixation rates of ~0.5 Tg-N yr$^{-1}$ is calculated for present-day conditions (Table 1). The increased soluble Fe inputs (~25%) relative to the STD simulations - although smaller compared to relative increases of N (~45%) and P (~50%) - tend to reduce the Fe-limitation in diazotrophs. Consequently, the reduced Fe-limitations make the extra atmospheric inputs of ON to the ocean more effective, overall decreasing the global nitrogen fixation rates in the model (Fig. 8a). Note, however, that the nitrogen fixation is a rather energy-expensive process that is known to be inhibited in the excess of ammonium, in particular. In the tropical Pacific Ocean, the nitrogen fixation rates for the ORG simulation are significantly more intense compared to STD (up to ~90%) but suppressed (up to ~40%) elsewhere (Fig. 8b). For example, in the Indonesian throughflow and the eastern tropical Atlantic along central Africa, significantly reduced rates (more than 90%) are calculated for the ORG simulation. The lowered nitrogen fixation rates in these regions are mainly due to the additional ON deposition in the ocean. Nitrogen fixation is also decreased in the tropical Atlantic Ocean. On the other hand, increased soluble Fe inputs to the tropical

Pacific (Fig. S3d), partially lower the Fe limitation of diazotrophs and increase nitrogen fixation in these remote oceanic regions. Overall, compared to the STD, the present-day global nitrogen fixation rate for the ORG simulation leads to a net decrease by only ~0.4% (Table 1).

Primary production increases almost in all ocean basins for the ORG simulation (Fig. 8d), except some parts of the Subpolar Pacific Ocean. In particular, higher rates are calculated in the subpolar Atlantic Ocean (up to 15%). In the N-limited oceanic regions, the increased atmospheric nitrogen deposition (Fig. S3b) directly increases the production rates (Fig. 8d). Such a case is the western subtropical North Pacific, where atmospheric N deposition supports an extra production of up to 15%. The production rates are also increased in the subtropical South Pacific and Atlantic Oceans up to nearly 20%. In total, the primary production increased from ~46.7 Pg-C yr$^{-1}$ for the STD to ~47.8 Pg-C yr$^{-1}$ for the ORG (Table 1). Figure 8d points out to regions in the Pacific where production decreased. For the North Pacific, this represents the same mechanism as described above for the differences in primary production rates between PAST and PRESENT (Sect. 3.3). For the ORG simulation, the increased iron input in the Pacific subpolar gyre increases diatom production leading to higher consumption of nitrate (Fig. S10b). Subsequent transport of nitrogen diminished waters further to the south cause a decrease in productivity further south. The boundary between decreased and increased bands matches the sharp transition from iron limitation to nitrogen limitation (Figs. 4a,c). Increased iron input south of the boundary (i.e., where Fe limits) stimulates the production and diminishes nitrogen. However, the advective mixing of the N-diminished waters with waters further north decreases productivity north of the boundary (i.e., where N limits). Overall, the result is the dipole pattern as demonstrated in Fig. 8d.

## 5 Discussion

The main focus of this study is to investigate the effect of nutrient deposition on oceanic primary production. Hence, the presented experiments did not account for the impact of future climate change which could interact or may even mask the effect of changed atmospheric deposition fluxes considered here. Consequently, the here found effects are subject to some uncertainties related to the potential interaction with climate change. For example, climate-induced changes in the global wind system may not only alter atmospheric pathways for nutrients but also impact on oceanic up- and down-welling. Thus, shifts in the seasonal position of trade winds will likewise force shifts in the position of open-ocean and coastal upwelling. These regions are usually nutrient-rich and not particularly sensitive to varying atmospheric nutrient inputs. Displacements of these upwelling positions as a result of climate change can increase the sensitivity to external nutrient inputs in regions formerly impacted by upwelling.

Several studies have demonstrated that mid to high latitude areas, such as the North Atlantic and the Arctic, will be more stratified in a future warmer climate (Bindoff et al., 2019; Fu et al., 2016; Gröger et al., 2013; Sein et al., 2018; Steinacher et al., 2010), with negative feedback on vertical mixing and marine primary production due to reduced upward transport of nutrients into the photic zone. Accordingly, primary production in these regions will probably be more sensitive to changed atmospheric deposition rates in the future climate. Our results overall imply only marginal effects in polar regions like the

Arctic Ocean. This is certainly robust under the present climate when marine productivity is limited by temperature and sea-ice reducing light conditions in these regions. However, there is a large agreement that climate change will be most severe in the high latitudes, with strong increases in the water temperatures and substantially diminished sea ice cover in the Arctic (Collins et al., 2013). Temperature and sea-ice related light limitation will likely become less important in a future climate in this region and thus, more nutrients will be recycled in the polar region and less exported equatorward. Consequently, changes in atmospheric transport and deposition of the bioavailable nutrients may play a larger role in the future climate than today, especially under the high-emission climate scenarios. An example can be seen in the high latitudes of the Southern Ocean around Antarctica where the major amount of surplus DFe is deposited in our PAST and FUTURE experiments. As expected, the additional DFe availability has nearly no effect on productivity as convective mixing and extremely low water temperatures maintain sufficient nutrients and support low productivity under the present-day climate. This may, however, change with altered oceanographic conditions under a future warmer climate. In the northern hemisphere, namely the northernmost Pacific, known as an HNLC region where iron is the limiting factor, the increased supply of DFe clearly stimulates marine productivity in the PRESENT and FUTURE periods compared to PAST. However, this increase in productivity is likely overestimated since our experiments lack climate-induced changes in future stratification which would reduce the nutrient supply from the deep ocean.

The impact of atmospheric organic nutrients on the global oceanic productivity turns out as high (~1 Pg-C yr$^{-1}$; Table 1) as the increase in the present-day primary production since preindustrial times when only inorganic nutrients' supply is accounted for. However, all changes in nutrient deposition fluxes here accounted for, are solely driven by changes in the anthropogenic and biomass burning emissions, along with the changes in insoluble to soluble conversions rates due to atmospheric processing. Thus, the atmospheric deposition fields used in this study did not account for any changes in dust and bioaerosol emissions. Instead, they were kept constant to the present-day atmosphere (i.e., the year 2010), although several studies suggest that dust fluxes may be sensitive to climate change and the land-use changes (e.g., Ginoux et al., 2012; Mahowald et al., 2010; Prospero and Lamb, 2003), and thus could be an important driver of the atmospheric nutrient cycles.

## 6 Summary and conclusions

This study presents the implementation of state-of-the-art monthly mean atmospheric deposition fields in the global biogeochemistry model PISCES. The model is here run in offline modus, forced by dynamical physical outputs from the physical ocean model NEMO. The newly coupled atmospheric deposition fields considered for this work are all calculated based on a detailed representation of emissions of natural and combustion nutrient-containing aerosols, detailed atmospheric gas- and aqueous-phase chemical schemes, and mineral dissolution processes due to atmospheric acidity and organic ligands. Another feature tested in the present study is the contribution of organic components to the atmospheric inputs to the global ocean. Moreover, to effectively isolate here the impact of atmospheric deposition on the marine biogeochemistry parameters, the atmospheric $CO_2$ mixing ratio is set to the preindustrial values for all simulations.

For the present day, ~40 Tg-N yr$^{-1}$, ~0.28 Tg-Fe yr$^{-1}$, and ~0.10 Tg-P yr$^{-1}$ of nitrogen, iron, and phosphorus atmospheric inputs to the global ocean, accounting for their inorganic fractions, are considered in PISCES. This results in a global nitrogen-fixation rate of ~112 Tg-N yr$^{-1}$ and an integrated primary production of roughly 47 Pg-C yr$^{-1}$. Compared to present-day conditions, the lower preindustrial atmospheric nutrient inputs to the ocean result in a weakened primary production of ~2%

globally. The decrease in oceanic productivity is supported by the preindustrial decrease in the soluble iron inputs, resulting from changes in combustion sources and the atmospheric processing of mineral aerosols along with the substantial decrease in atmospheric anthropogenic nitrogen inputs. The projected changes in air pollutants under the RCP8.5 emission scenario also result in a modest decrease in marine productivity compared to modern times. Global nitrogen-fixation rates present here a marginal variability, although some notable decreases are calculated for the modern subtropical Pacific and Atlantic gyres.

This work asserts the importance of an explicit representation of the atmospheric nutrients in the context of biogeochemistry modeling, providing also a first assessment of the contribution of another source of atmospheric nutrients than inorganics and, thus, the potential importance of organic nutrients on oceanic productivity. Overall, our main conclusions can be summarized as:

    1)   An overall low impact of atmospheric nutrient deposition scenarios on total marine primary production on a global
scale. This is because much of modern productivity is driven by nutrients already recycled in the euphotic zone or by nutrient import from the deep ocean (such as in upwelling regions). Additionally, atmospheric transport appears rather important, as a significant part of nutrient deposition takes place in the northern high latitudes, where light conditions and temperature further limit productivity. Accordingly, even substantial reductions of nitrogen, phosphorus, and iron inputs, ranging between 36 and 51% during the preindustrial period, result in an only modest decline of primary
production of about 3%.

    2)   Substantial local productivity changes of up to 20% are found in regions limited by nutrients. The strongest sensitivity against atmospheric nutrients is found for the oligotrophic subtropical gyres of the North Atlantic and the North Pacific, where good light conditions and warm temperatures together with low nutrient concentrations predominate. Additional atmospheric nutrient input to these regions immediately results in production by increasing the biogenic
turnover.

    3)   The North Pacific turns out to be the most sensitive to iron deposition. For the preindustrial period, the lowered input of iron to this region leads to a strong decline of siliceous diatom production leading to an enrichment of silicate, nitrogen, and phosphorus. In turn, this leads to enhanced equatorward transport of nutrients resulting in elevated production rates of calcareous nanophytoplankton further southeast.

4)   The North Pacific appears more sensitive to external nutrient atmospheric deposition compared to other oceanic regions mainly due to two reasons: the strongest deposition changes take place in the northern mid to high latitudes, and that compared to the Southern Ocean and the North Atlantic, the exchange with cold and nutrient-enriched polar waters is limited by land by the shallow Bering Strait and the Aleutian arc. By contrast, the southern high latitude ocean contains a large amount of unutilized nutrients that are advected further north (to mid-latitudes) making this

region more robust against changes in external nutrient input. In agreement, however, with observational evidence from WOA, PISCES exhibits a widespread surplus of nitrogen compared to phosphorus and with respect to the Redfield ratio. Therefore, the applied changes in phosphorus inputs have nearly no impact on primary production in the model. This applies even to the warm water regions, where reductions in atmospheric iron supply limit nitrogen fixation by diazotrophs in both PAST and FUTURE periods.

5) Finally, the effect of atmospheric organic nutrient deposition fluxes on the global primary production is calculated roughly as strong as the effect of the present-day increased emissions and atmospheric processing on the oceanic biogeochemistry since preindustrial times when only the inorganic fraction is considered in the model ($\sim$1 Pg-C yr$^{-1}$). Note that based on the Krishnamurthy et al. (2009) model estimations, an atmospheric $pCO_2$ declined of about 2.2 ppm due to the modern era's iron and nitrogen inputs to the global ocean can be supported. Accordingly, the here calculated increase in primary production related to the input of organic nutrients could correspond to a respective decrease in atmospheric $pCO_2$ of $\sim$1.6 ppm. Although the overall impact of atmospheric organic nutrient deposition on a global scale is rather low, some stronger changes in regional oceanic productivity are clearly demonstrated in the oligotrophic subtropical gyres.

**Data availability.** Atmospheric nutrient deposition data used for this study are available at Zenodo (doi: 10.5281/zenodo.4017209).

**Author contribution.** SM prepared the atmospheric input fields, performed the simulations, and the model evaluation. MG
prepared the model forcing data. SM and MG wrote the manuscript. All authors contributed to the manuscript preparation.

**Acknowledgments.** Stelios Myriokefalitakis acknowledges financial support for this research from the European Union's Horizon 2020 research and innovation programme under the Marie Skłodowska-Curie grant agreement no. 705652 – ODEON. Matthias Gröger and Jenny Hieronymus acknowledge support from the European Commission's Horizon 2020 Framework
Programme, under Grant Agreement number 641816, the "Coordinated Research in Earth Systems and Climate: Experiments, kNowledge, Dissemination and Outreach (CRESCENDO)". Stelios Myriokefalitakis acknowledges support from the Joint Group of Experts on the Scientific Aspects of Marine Environmental Protection (GESAMP; http://www.gesamp.org/), Working Group 38, the Atmospheric Input of Chemicals to the Ocean. The authors thank Alessandro Tagliabue for providing the dissolved iron oceanic concentration dataset. Model simulations were carried out on the Bi cluster operated by the Swedish
National Supercomputer Centre (http://www.nsc.liu.se/) and data processing is supported by computational time granted by the Greek Research and Technology Network (GRNET) in the National HPC facility ARIS (https://hpc.grnet.gr/) under the project PR006002-ADIOS. The authors thank two anonymous reviewers for their comments that significantly helped to improve the final manuscript. The publication of this work was financed by the internal grant program of the National Observatory of Athens "Atmospheric deposition impacts on the ocean system" (number 5065).

**Competing interests.** The authors declare that they have no conflict of interest.

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

**Tables and Figures**

Table 1: Nutrients (N, Fe, P) atmospheric inputs (Tg yr$^{-1}$) considered in PISCES, nitrogen fixation (Tg-N yr$^{-1}$), and primary production (Pg-C yr$^{-1}$) as calculated by the STD and ORG simulations, for PAST (1851–1870 average), PRESENT (2001–2020 average) and FUTURE (2081–2100 average).

| | Atmospheric Input Tg yr$^{-1}$ | | | Nitrogen Fixation Tg-N yr$^{-1}$ | | Primary Production Pg-C yr$^{-1}$ | |
|---|---|---|---|---|---|---|---|
| | Nutrient | STD | ORG | STD | ORG | STD | ORG |
| PAST | DN | 19.74 | 34.47 | 111.62 | 112.22 | 45.46 | 46.43 |
| | DFe | 0.18 | 0.23 | | | | |
| | DP | 0.06 | 0.11 | | | | |
| PRESENT | DN | 40.01 | 58.01 | 111.87 | 111.41 | 46.65 | 47.75 |
| | DFe | 0.28 | 0.35 | | | | |
| | DP | 0.10 | 0.15 | | | | |
| FUTURE | DN | 34.62 | 51.08 | 110.75 | 110.65 | 46.36 | 47.42 |
| | DFe | 0.24 | 0.30 | | | | |
| | DP | 0.08 | 0.13 | | | | |

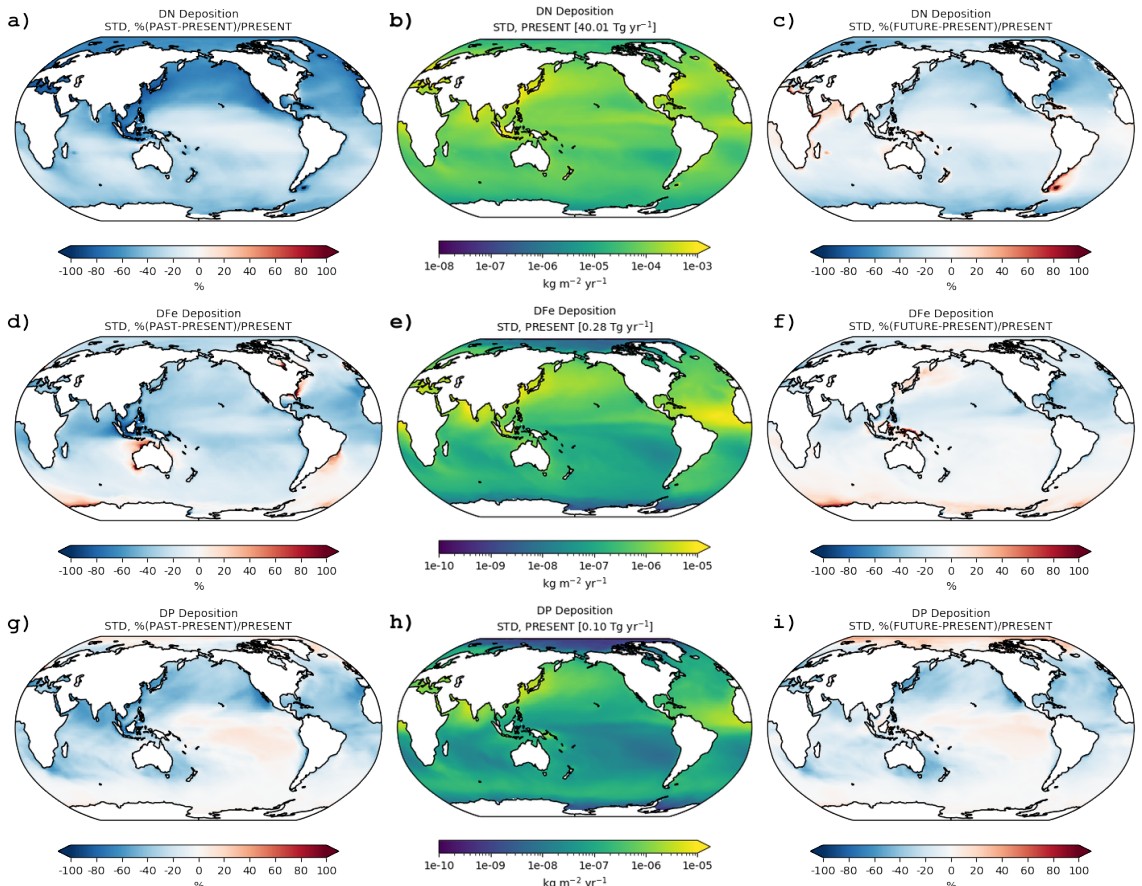

**Figure 1: Atmospheric deposition fluxes into the ocean (kg m⁻² yr⁻¹) of dissolved nitrogen (b), iron (e), and phosphorus (h) considered by the model for PRESENT (middle column) for the STD simulation, and the respective relative differences (%) to PAST (left column) and FUTURE (right column).**

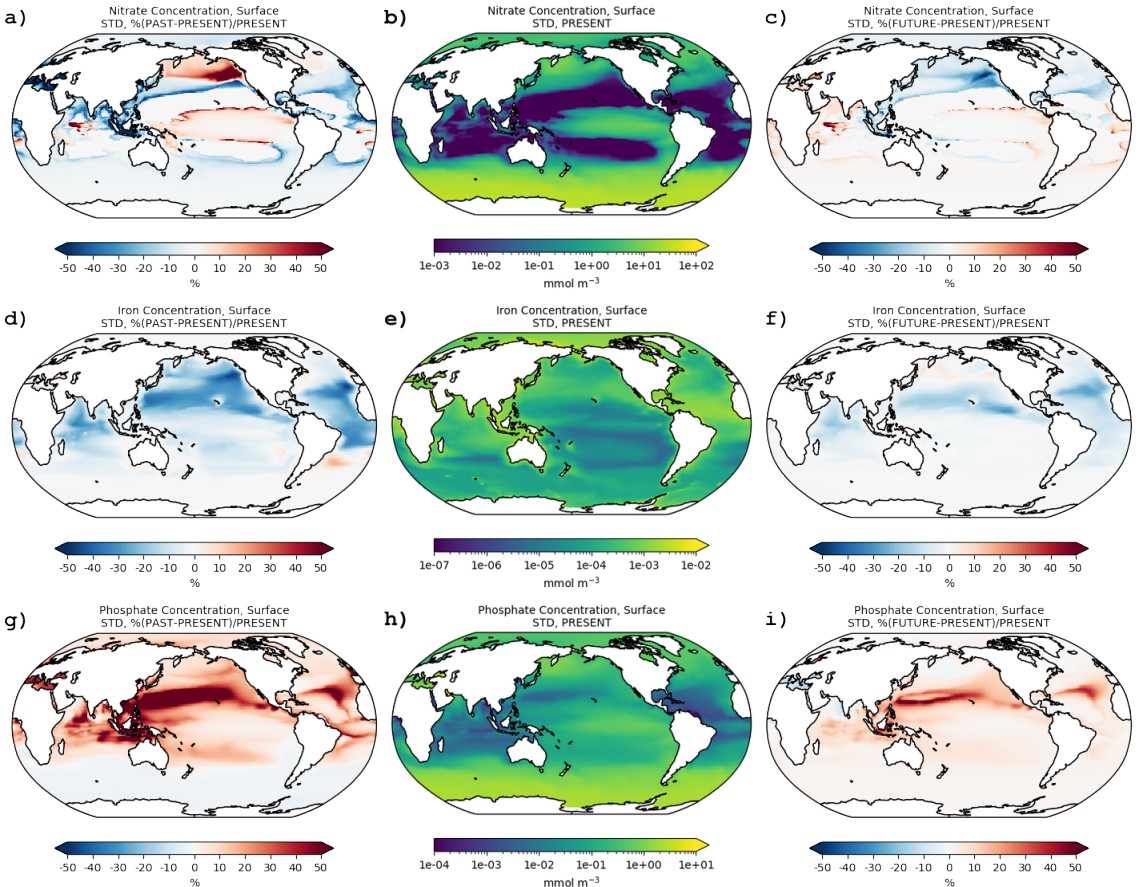

**Figure 2: Surface oceanic concentrations (mmol m⁻³) of nitrate (b), iron (e), and phosphate (h) as calculated by the model for PRESENT (middle column) for the STD simulation, and the respective relative differences (%) to PAST (left column) and FUTURE (right column).**

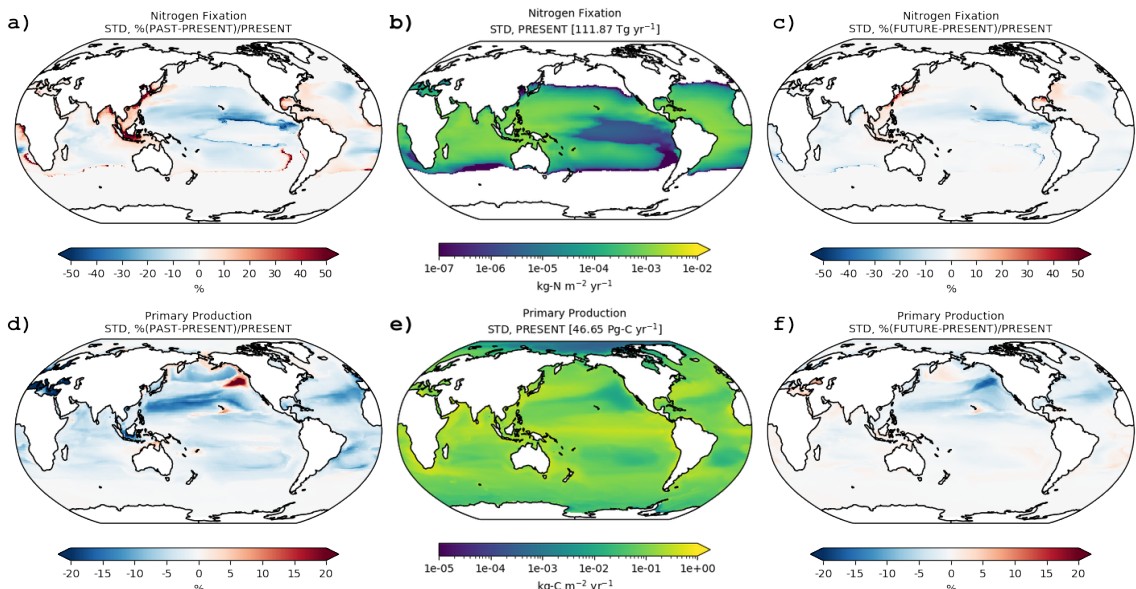

**Figure 3: Nitrogen fixation (kg-N m⁻² yr⁻¹) and primary production (kg-C m⁻² yr⁻¹) rates as calculated by the model for PRESENT (b,e) for the STD simulation, and the respective relative differences (%) to PAST (a,d) and FUTURE (c,f).**

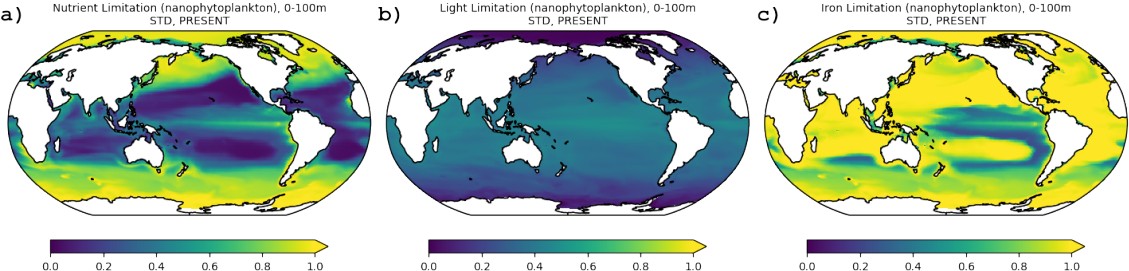

**Figure 4: Limitation for nanophytoplankton production by nutrients (N and P; a), light (b), and iron (c). Low values indicate high limitation imposed by the respective property.**

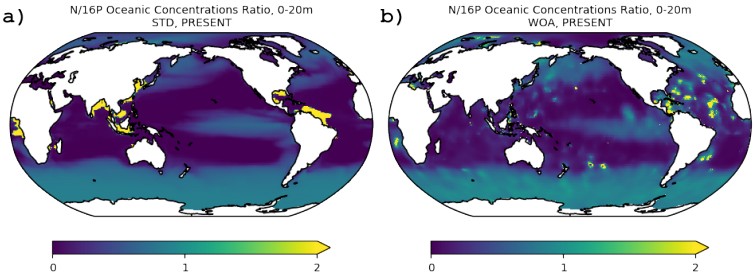

5    **Figure 5: Molar oceanic ratios N:16P averaged in the upper 20m for PRESENT, as calculated by the model (a) for the STD simulation, and based on World Ocean Atlas (WOA; Garcia et al., 2010b) data (b). Values >1.0 denote overshoot of N vs P relative to the Redfield ratio (C:N:P =122:16:1); blue areas indicate a surplus of P or deficiency of N.**

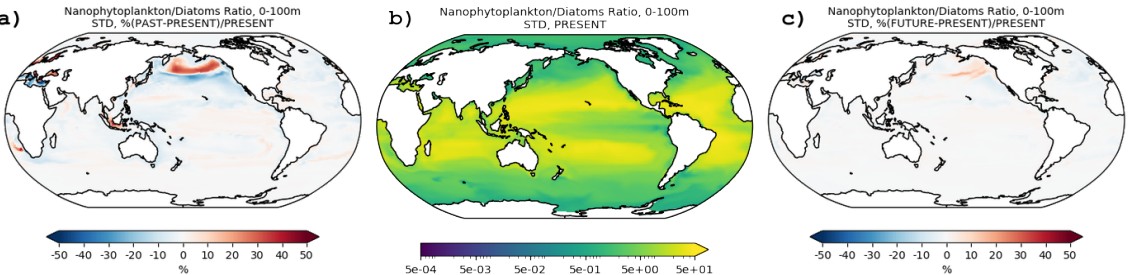

10   **Figure 6: Nanophytoplankton to diatoms oceanic concentrations ratio averaged in the upper 100m for PRESENT (b) for the STD simulation, and the relative changes to PAST (a) and FUTURE (b).**

**a)**

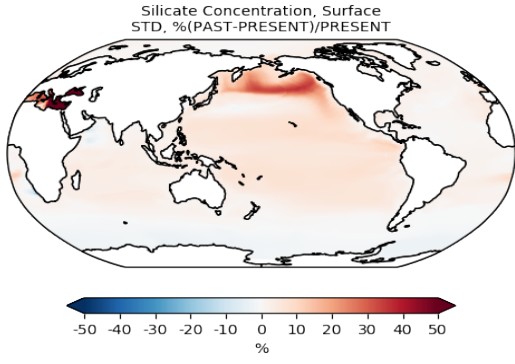

**b)**

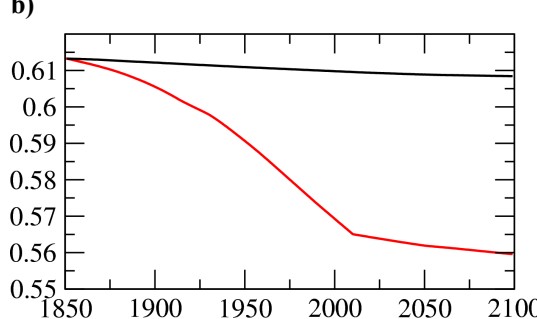

Figure 7: a) PAST to PRESENT relative differences (%) of silicate surface oceanic concentrations of as calculated by the model for the STD simulation; b) Seawater concentration ratio of nanophytoplankton to diatoms in the upper 20m, averaged over the NW Pacific (east of 200°E and north of 40°N). The red line indicates primary production rates for the STD simulation and the black line for the CTRL simulation, respectively.

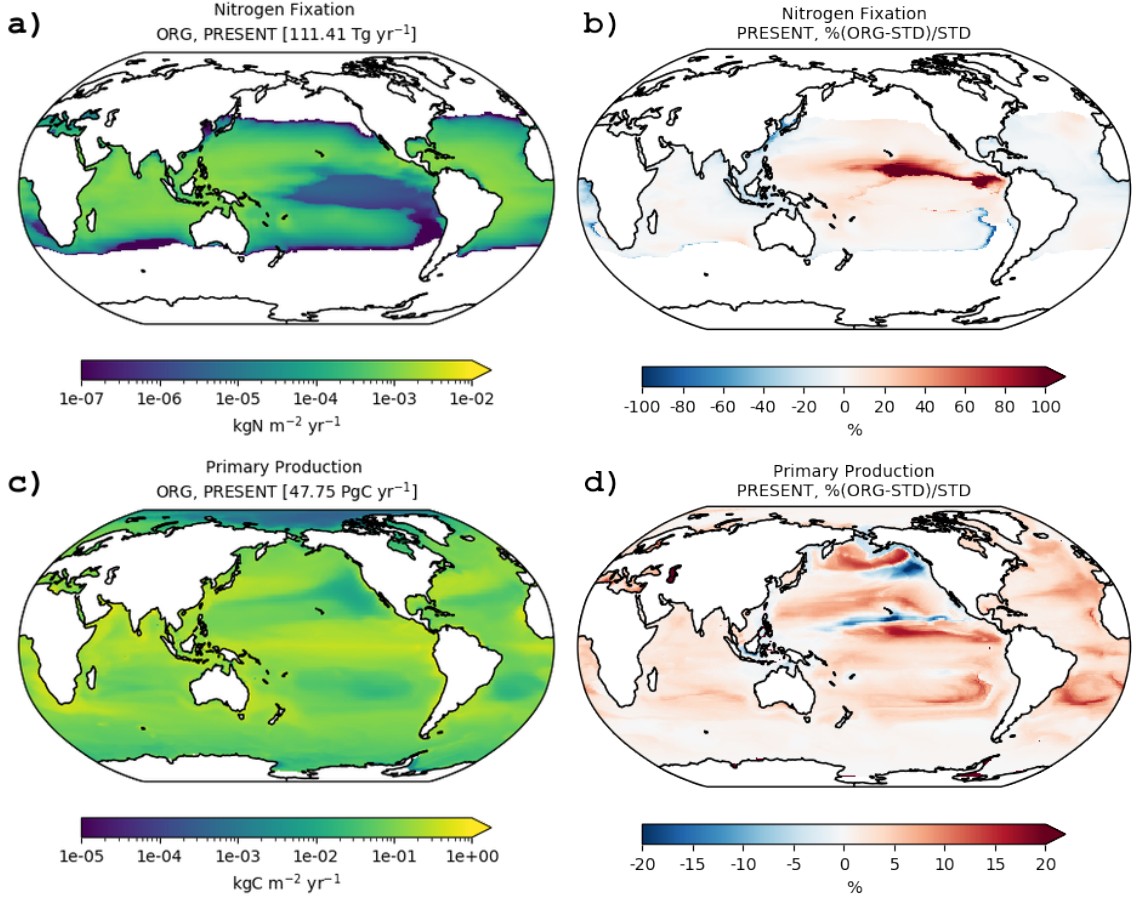

**Figure 8: Nitrogen fixation (kg-N m$^{-2}$ yr$^{-1}$) and primary production (kg-C m$^{-2}$ yr$^{-1}$) rates as calculated by the model (a,c) for the ORG simulation for PRESENT, and the respective relative (%) differences (b,d) to the STD simulation.**