# Peer review of "An explicit estimate of the atmospheric nutrient impact on global oceanic productivity"

_Ocean Science, 2020_

## Referee Comment (RC1) · Anonymous Referee #1 · 22 May 2020

Myriokefalitakis and colleagues run transient offline simulations with the PISCE ocean ecosystem model forced with output for N, Fe, and P deposition from state-of-the-art atmospheric chemistry models. The forcing fields are interpolated from the available deposition fields for the preindustrial ($\sim$1850) to present to end of 21st century. Changes in other drivers such as increasing atmospheric CO2 and 21st century global warming are not explicitly considered.

The authors investigate changes in surface nutrient concentrations in response to generally strongly increasing deposition over the industrial period and decreasing deposition over this century (their Fig 1). The authors compare the differences between three periods - the past (1851-1870), present (2001-2020) and future (2081-2100) (Fig 2 and 6) and between simulations with and without including organic forms of deposited nutrients (Fig. 8). The authors also compare simulated surface nutrient and productivity fields with available observational data (Fig 3-5 and 7).

The authors find that changes in inorganic and organic deposition of nutrients have a small influence on global marine productivity, but with regionally stronger effects. The value of the study is in using detailed deposition fluxes for inorganic and organic forms to assess their roles in influencing surface nutrient concentrations, primary productivity, and nitrogen fixation as simulated by a 3-d ecosystem model. This is a new and valuable contribution.

I have a few points for potential improvements:

(1) The current model setup is not clear to the reviewer and section 2.1 on the model setup needs to be improved for clarity. I guess the PISCES ecosystem model is run off-line with physical transport, T, and S fields taken from a physical ocean only simulation. I guess the ocean only simulation is forced with repeated time-varying surface temperature and salinity fields taken from observational data for years 1948 to 2009. This would include a strong warming over the 61-yr period, followed by a rapid "cooling" and again 61 years of warming, etc. In my opinion section 2.1 needs a rewrite and should be structured in a much better way. The description of the setup of the model used to get the forcing fields (circulation , T, S?) to drive PISCES and the spin up and drift of this part of the model chain should be clearly separated from the setup of PISCES.

(2) Simulated changes in nutrient concentrations and productivity are relatively modest. The question arises whether the difference in simulated surface nutrient concentrations (Fig. 2) and productivity (Fig. 6) between PI, present and future periods are only due to differences in the deposition fields as implied by the manuscript or also influenced by other factors. Namely, model drift and, potentially also very important, differences in the physical fields used to force PISCES between the three period of interests could be responsible for some of the differences. I am not sure and I may be wrong, but I have

the impression that the physical fields used to force PISCES are taken from different nominal years of the ocean only simulations and therefore different ocean circulation fields could explain part of the simulated differences in nutrient concentrations and productivity. I also miss the mentioning of a true control run with constant deposition and identical forcing as the standard transient runs with time-varying deposition. This would allow the authors to correct for drifts and changes related to physical forcing. As modelled changes are relatively small, this appears particularly important.

(3) a) Further, I am wondering whether the four figures with 27 maps used to compare simulated with observed fields are really that relevant for this study. They distract from the other, very nice and important figures. Surface nutrient and productivity fields for PISCES have been compared with observations in earlier studies. These simulated fields result predominantly from physical transport of nutrients within the ocean and from the PISCES model itself, whereas the role of atmospheric deposition is rather marginal. The comparison in these four figures tells us, in my opinion, not much about the topic of this study – atmospheric nutrient deposition. They may be included in an appendix and the corresponding text can, in my view, be drastically shortened.

b) On the other hand, I miss some assessment how changes in deposition influence surface nutrients or productivity regionally. Is this due to local effects/deposition? Or is there an influence of ocean surface transport in bringing deposited material to other regions? I have not firm recommendation, but some analysis would be useful. I could imagine to correlate changes in deposition fields with changes in simulated fields or to run factorial simulations with deposition varying only in certain regions, though this may be too CPU expensive.

c) I miss a figure that shows time series of the prescribed transient evolution of globally averaged deposition of N, P, F, from 1850 to 2100. Potentially one could show in this figure also the share of inorganic and organic forms or of different sources. This figure should also include a time serie(s) representing the evolution of the applied physical forcing (e.g. global mean sea surface temp or SAT). In this way, the reader could quickly

understand how and what is varied in the simulations and this figure would complement the table and current fig. 1

Detailed comments

P1, l9: Immediately when reading the abstract one starts to wonder what kind of physical ocean model is used to power PISCES. Please clarify that PISCES is coupled offline (?) to a forced ocean only simulation.

P1,l9: Please mention how atmospheric CO2 and climate change is included.

P3, l29: Is it correct to say that Fe-containing combustion aerosols are mainly deposited in the Pacific and Southern Ocean? Or do you mean that combustion aerosols play a larger role (compared to dust) in these regions?

P5 l19-22: I do not understand what the authors want to say here. Is PISCES now fully coupled online to NEMO and the EC-Earth ESM or rather forced offline with an OMIP simulation? I think it should read: "The state-of-the art biogeochemistry PISCES model is here run "offline" with prescribed transport and T, and S fields (see sec. 2.1). The version of PISCES implemented within NEMO and the European Earth System Model EC-Earth is used in this study. PISCES simulates the . . .

P6,l5: please specify the physical output used to force PISCES.

P6,l6: which OMIP simulation?

P6, l6: Please specify how the 1948 to 2009 forcing is aligned in the spin-up of PISCES and the transient simulation from 1850 to 2100.

P6, l7: I guess this referes to the spin up for ocean model without PISCES? Please clarify.

P6, l8-l10: again – to which simulation or model does this initialization apply?

P6, l8-l10: I guess there are still substantial drifts in O2,N, Si, P and Alk after such a

short spinup of 300 years only. Please specific how large the drifts are.

P6, l8-10: Is there any initialization of DIC or DOM?

P6: what is the role of global warming/climate change in these simulations?

P6: Has a true control run with constant dust deposition and same physical forcing be applied for the 1850 to 2100 period? The drift in critical variables should be quantified.

P6, l29: I find the labeling of the first simulation as "CTRL" very misleading. For me this is the standard simulation with time-varying deposition forcing. Please select another name for this simulation.

P7, line 5: Here a second spin-up is mentioned. The structure of the section is confusion, switching between a first spin-up (of the ocean model), transient PISCES simulation, a second spin up (for PISCES?) and again transient simulation. Please streamline the structure of section 2.1

P7, line 7: No indication is given how this drift is quantified and for which period it holds and whether this is for the global average or for each horizontal grid cell. Please specify.

P6,p7: Is the physical forcing to PISCES identical for the three periods (past, present , future). If not, what would be the implication of differences in the physical forcing?

Section 2: A figure showing the time series of global mean deposition of P, N, and Fe would be really useful.

Section 2.2 A note how these atmospheric deposition fluxes compare with the riverine input would be helpful.

Section 2.2. A note how these atmospheric deposition fluxes compare with export (or new) production of P, Fe, N (as particulate and dissolved organic forms)

P11, l29: Is production limited by light as suggested here or also be Fe?

P17, line 8: Please state how PISCES is forced and what circulation fields are used

P17, l14-20: Part of this text should be mirrored in section 2 where the model setup is described. For example, the mentioning of salinity restoring comes somewhat at a surprise as it is not clear from section 2 that restoring boundary conditions are applied. Similarly, you talk about a prolongation of the OMIP simulation using an RCP8.5 scenario run. Again, this seems not to be described in the method section. Please provide a complete description of the model setup in section 2.

P20: It would be very useful for the community if the deposition files would be made available, e.g. as netcdf files, to the community. I miss a corresponding data availability statement.

---

## Referee Comment (RC2) · Anonymous Referee #2 · 3 Jun 2020

In this manuscript, Myriokefalitakis and coauthors quantify the regional and global effects of atmospheric deposition of nutrients to the ocean, consistently derived from a novel, comprehensive atmospheric chemistry model in conjunction with a framework for ocean biogeochemical cycling and marine productivity. Present day atmospheric deposition of the biologically-essential elements nitrogen (N), phosphorus (P), and iron (Fe) appears to be at a peak largely driven by anthropogenic activity, such as emissions from air and sea transport and land use changes (e.g. biomass burning), which has heterogeneously increased N, P, and Fe sources to the ocean since the preindustrial era. These sources are projected to decrease into the future. Despite significant changes in atmospheric nutrient deposition (around 2x more deposition in the present than the preindustrial, and 10-20% decrease in the future) primary production and

nitrogen fixation rates remained relatively consistent over the entire period, although regional changes are more significant. Inclusion of organic deposition, in addition to inorganic deposition, resulted in a similar magnitude of nutrient, primary production and nitrogen fixation anomalies compared to inorganic deposition alone. This is an interesting topic with a novel approach that merits investigation, and eventual publication in Ocean Sciences.

General points:

While the authors systematically validated their present day simulation against observations and described the effects of their new atmospheric nutrient deposition fields on surface ocean nutrient concentrations, as well as the rates of primary production and nitrogen fixation, I found that the quantity and organization of the material eclipsed crucial results, and that the depth of the analysis that was presented was somewhat limited.

Since the title emphasizes global oceanic productivity I was expecting significantly more discussion about the emergent rates of primary production and nitrogen fixation (currently ∼1 page combined). That biological productivity/nitrogen fixation is relatively stable at the global scale while more significant changes occur regionally implies a compensatory mechanism, which is not really explored. I was looking for more information supported by encompassing and generalizing diagnostics than the numerous supplied maps could provide.

- How does the ratio of the atmospheric supply of nutrients change regionally/globally

- How does the combination of these resources promote or inhibit production vs diazotrophy?

- Are phytoplankton (or diazotrophs) consuming critical resources "upstream" that inhibit "downstream" productivity via scarcity or changing nutrient ratios?

- Are unutilized nutrients (e.g. Southern Ocean iron in the future) transported away

from the surface to reemerge elsewhere and stimulate productivity remotely?

- Are there teleconnections associated with regions of enhanced export and enriched deep water nutrients upwelling elsewhere?

- What about silicate (Si) fluxes?

- How did the composition of phytoplankton functional groups (diatoms vs other phytoplankton) change?

- Is production limited by a top-down grazing pressure, or a bottom-up resource limitation?

Some of these issues were touched upon when explaining the counterintuitive higher oceanic P concentrations simulated for the preindustrial era despite lower P deposition, which I found really interesting. There are many "moving parts" associated with this study that some idealized experiments might help disentangle the mechanistic role of atmospheric nutrient deposition on ocean biogeochemistry and production. Perhaps substitution experiments with the newly derived N, P, or Fe deposition singly swapped with remaining "standard" PISCES inputs (or combinations of two substituted out of three).

I appreciated the signposting of the manuscript structure at the end of the introduction, but I thought the paper would benefit from having separate "results" and "discussion" sections, with integrative diagnostics in the former, and more emphasis on explaining the changes in emergent ocean properties and comparisons with previous studies such as Krishnamurthy in the latter. At the moment, the key messages are very much buried within the qualitative/semi-quantitative description of the results. I also think that the model-data comparisons, although reassuring, interrupted the flow of presentation. One could create a new section on model validation, but I would recommend moving the material and figures to Supplementary information. One comparison that may have been really interesting to include is a comparison of the "CTRL" run to a vanilla PISCES

simulation, with the standard atmospheric nutrient fluxes.

Specific comments:

P3, line 20: "no-linearly" typo.

P4, line 13: "It has also been suggested..." citation needed, unless it's Krishnamurthy et al. (2010) in which case rephrase for clarity.

P6, line 7: "five iterations" I think it would be more precise to say you ran the model for 300 years, repeating the 60 year physical forcing five times. Five iterations could technically imply a spin up of 5x2700s.

P6, line 9: Which versions of WOA and GLODAP did you use (if not WOA2013 and GLODAPv2). How was DIC initialized? Also from GLODAP?

P6, line 25: "no extra optimizations for the iron scavenging parameters" The specific iron cycle configuration is of critical importance to understanding the effect of changing iron input, please can you give more details about this? Did you use particle dependent scavenging? How is organic ligand complexation parameterized (constant or variable ligand concentration)?

P7, line 1-12: Timeseries of the model nutrient sources would clarify how the experiment was run i.e. I think you did one transient run 1651-2100 and analyze the nutrient concentrations/ecosystem response at three 20-year average periods. In addition, it would be great to show the temporal evolution of globally/regionally-averaged nutrient concentrations and emergent diagnostics during this run. This got me thinking about whether the "present day" actually represents the peak in atmospheric nutrient deposition, or if that was earlier (70's, 80's or 90's)? There was no real justification for choosing the 2001-2020 average given.

P7, line 12: Will these datasets be available online for ESM groups to experiment with?

P11, line 18: How do dust and aerosol emissions, that are not considered, vary over

the time period in question? I think this is touched upon in the "summary"....

P11, line 26: "cooler water temperatures..." caused also by high latitude upwelling?

P12, line 5: "leads to more efficient export..." N supply may certainly lead to increased rates of export in nitrate-limited oligotrophic regions, but if the nutrients are drawn down to the same low levels, for example in the gyres, is the export actually more efficient?

P12, line 33: It would be an even more convincing model-data comparison if the authors took advantage of the extensive GEOTRACES iron dataset (https://www.geotraces.org/geotraces-intermediate-data-product-2017/) with ∼6 years of cross calibrated additional data from a concerted international effort.

P13, line 21-25: side note about Redfield ratios might be better placed with the model set up.

P14, line 14: Why does nitrogen fixation decrease?

P14, line 22: "with the projected decrease of the global inorganic nitrogen and iron inputs..." Nitrogen fixation should be promoted by lower N:P ratio (i.e. decreased N and increased P) so is the lower fixation rate due to iron limitation? Is it possible to show maps of resource limitation from the model for phytoplankton/diazotrophs, e.g. the limiting terms in Equation 6 in Aumont et al. (2015)?

P15, line 32: "all dissolved organic matter is assumed to be instantaneously remineralized..." I think this is incorrect. Equation 32 in Aumont et al. (2015) shows how dissolved inorganic matter (for carbon and other species related by fixed Redfield ratios) is separately modeled as a pool supplied by phytoplankton and zooplankton exudation and remineralized aerobically or anaerobically by bacteria over a timescale of the order of months to years.

P16, line 19: "in contrast to the rather balanced nitrogen fixation rates..." a 2% change in primary production also sounds rather balanced to me.

P16, line 20-28: 15-20% increases occur relatively widely in the ocean, so what causes the counterbalancing decline in productivity? Why are the decreases confined to these bands in the Pacific?

P17, line 13-20: Salinity restoring and mixed layer dynamics was never mentioned in the main text, so surprised to see it prominently in the "Summary" section.

Figures: It would be preferable to use a perceptually uniform color palette for the CTRL maps, as opposed to the rainbow/jet colormap currently shown (see here for details: https://blogs.egu.eu/divisions/gd/2017/08/23/the-rainbow-colour-map/, not to mention the accessibility issue surrounding red/green vision deficiency).

---

## Author Comment (AC1) · 31 Aug 2020

We thank the reviewer for the careful reading of the manuscript and the fruitful comments. Please find below our point-by-point replies:

**I. General Comments**

- GC1. The current model setup is not clear to the reviewer and section 2.1 on the model setup needs to be improved for clarity. I guess the PISCES ecosystem model is run off-line with physical transport, T, and S fields taken from a physical ocean only simulation. I guess the ocean only simulation is forced with repeated time-varying surface temperature and salinity fields taken from observational data for years 1948 to 2009. In my opinion section 2.1 needs a rewrite and should be structured in a much better way. The description of the setup of the model used to get the forcing fields (circulation, T, S?) to drive PISCES and the spin up and drift of this part of the model chain should be clearly separated from the setup of PISCES.
  - PISCES model is run off-line for this work, with the physical transport, temperature, and salinity fields taken from a physical ocean NEMO simulation. This NEMO simulation was driven five times (310 years) for the years 1948 to 2009 (Skyllas et al., 2019). For all PISCES offline simulations, we used the same 1-year physical ocean forcing. This one year was calculated as a multi-year daily mean over the 5th iteration 1948-2009 of the 310-year NEMO simulation. This way we removed interannual variability and any long-term trend in the physical forcing but conserved the full seasonal cycle on a daily basis. To avoid confusion, we rewrote Sect. 2 adding further analyses by including separately a description of the forcing data (Sect. 2.1.1) and a description of the PISCES model setup (now Sect. 2.1.2):

**"2.1.1 Physical Ocean forcing**

The dynamical physical outputs used to force PISCES for this study are produced by the physical ocean model NEMO, following the protocol of the OMIP simulation (Ocean Modelling Intercomparison Project; Orr et al., 2017). OMIP aims at harmonizing forcing fields of boundary conditions, as well as validation and analysis procedures among different ocean models. Atmospheric forcing fields are from the CORE II forcing (Coordinated Ocean-ice Reference Experiments - Phase II; Large & Yeager, 2009). CORE II provides a 62-year interannual forcing for the period 1948-2009. The physical model is initialized with gridded observational data from the World Ocean Atlas 2013 and then ran for 310 years by repeating the 62-year CORE II forcing. The necessary physical variables to force the offline biogeochemical model PISCES (see Table S1) are taken from the last 62-year iteration. To avoid, however, any long-term trends from spin up, the multi-year (1948-2009; i.e., the 5th iteration of the 310-year run) mean of daily forcing fields was calculated. The resulting mean 1-year forcing thus contains the mean seasonal cycle and was (repeatedly) applied to drive all simulations with the biogeochemical PISCES offline model. All biogeochemical simulations were initialized and forced with the same physical fields from the average 1-year forcing derived from the OMIP run. Thus, all the PISCES offline simulations are drift-free in physical variables. More details of the OMIP protocol can be found in Orr et al. (2017) and a first validation of the OMIP run is provided by Skyllas et al. (2019)."

| Water flux into seawater            | $kg/m^2/s$ |
|-------------------------------------|------------|
| Mixed layer depth                   | m          |
| Surface net downward shortwave flux | $W/m^2$    |
| Wind speed                          | m/s        |
| Ice concentration                   | %          |

**"Table S1. Physical forcing fields provided at a daily time step.**

| Water flux due to freezing/melting                      | kg/m²/s    |
|---------------------------------------------------------|------------|
| Tracer diffusive fluxes along the bottom boundary layer | $m^3/s$    |
| River runoff                                            | $kg/m^2/s$ |
| Ocean vertical salt diffusivity                         | $m^2/s$    |
| Horizontal divergence transport                         | 1/s        |
| Seawater salinity                                       | g/kg       |
| Seawater potential temperature                          | °C         |
| Effective ocean transports                              | $m^3/s$    |

- GC2. Simulated changes in nutrient concentrations and productivity are relatively modest. The question arises whether the difference in simulated surface nutrient concentrations (Fig. 2) and productivity (Fig. 6) between PI, present and future periods are only due to differences in the deposition fields as implied by the manuscript or also influenced by other factors. Namely, model drift and, potentially also very important, differences in the physical fields used to force PISCES between the three period of interests could be responsible for some of the differences. I am not sure and I may be wrong, but I have the impression that the physical fields used to force PISCES are taken from different nominal years of the ocean only simulations and therefore different ocean circulation fields could explain part of the simulated differences in nutrient concentrations and productivity. I also miss the mentioning of a true control run with constant deposition and identical forcing as the standard transient runs with time-varying deposition. This would allow the authors to correct for drifts and changes related to physical forcing. As modelled changes are relatively small, this appears particularly important.
  - We thank the reviewer for attracting our attention to this issue. Indeed, all simulations of this work were performed with the same forcing data. The only difference among the simulations was the atmospheric nutrient inputs to the ocean; thus, the differences in oceanic surface concentrations and productivity between PAST, PRESENT, and FUTURE periods are only due to the respective deposition fields considered by PISCES. We agree with the reviewer that a true control run that would correspond to a simulation with constant PI deposition and identical forcing as the transient simulation with time-varying deposition presented in the paper would be the appropriate way to show that the model drift in our simulations is low. For this, we now include a control run (i.e., with constant preindustrial deposition and identical forcing as the standard transient runs) that used for the drift correction of model results, i.e., the nutrient concentrations and marine productivity fields that are presented in the manuscript. The model drifts are calculated here using the linear detrend operator of the CDO (climate data operators) software. Although the impact is minimal thus without changing at all our conclusion, we have also updated all figures based on the resulted drift corrections and we added the following part in the new Sect. 2.1.2, i.e., "To account, however, for potential drifts, a control simulation as for STD but using only preindustrial (i.e., the year 1850) atmospheric nutrients (N, P, and Fe) inputs into the global ocean is also performed. For example, Fig. S1 demonstrates that for the main ocean basins the drift is minimal after 1850 and clearly below the signal imposed by the altered nutrient deposition. This holds even for the Southern Ocean where the impact of atmospheric deposition is typically weak due to the absence of neighbored emission sources. Nevertheless, all model results presented in this work have been adjusted by subtracting the drift of the control run from STD."

Figure S1: Area averaged annual mean primary production in mole-C m-3 s-1 for the main ocean basins. Red lines indicated primary production rates for the STD simulation and black lines the CTRL simulation, respectively.

- GC3. Further, I am wondering whether the four figures with 27 maps used to compare simulated with observed fields are really that relevant for this study. They distract from the other, very nice and important figures. Surface nutrient and productivity fields for PISCES have been compared with observations in earlier studies. These simulated fields result predominantly from physical transport of nutrients within the ocean and from the PISCES model itself, whereas the role of atmospheric deposition is rather marginal. The comparison in these four figures tells us, in my opinion, not much about the topic of this study atmospheric nutrient deposition. They may be included in an appendix and the corresponding text can, in my view, be drastically shortened.
  - We agree with the reviewer that the model evaluation part may distract the reader from the main conclusions of this work. For this, we now moved all model evaluation figures and the respective discussion to the supplement.
- GC4. On the other hand, I miss some assessment how changes in deposition influence surface nutrients or productivity regionally. Is this due to local effects/deposition? Or is there an influence of ocean surface transport in bringing deposited material to other regions? I have not firm recommendation, but some analysis would be useful. I could imagine to correlate changes in deposition fields with changes in simulated fields or to run factorial simulations with deposition varying only in certain regions, though this may be too CPU expensive.
  - Indeed, to really distinguish the local depositional forcing and the second-order effect of advection of unutilized nutrients on the local productivity would require several additional sensitivity simulations, not currently feasible with available resources. However, we note that in areas where a specific nutrient is not limiting, advection to remote places is highly likely. For the revised version, further analysis has been carried out to investigate and demonstrate this for e.g., the North Pacific in the new section 3.2.1 as well as in the new Discussion Section (i.e., Sect. 5 in the new version of the manuscript). Besides, an additional sensitivity simulation was performed with a constant P deposition, which demonstrated the importance of iron and nitrate compared to phosphorous.

- GC5. I miss a figure that shows time series of the prescribed transient evolution of globally averaged deposition of N, P, Fe, from 1850 to 2100. Potentially one could show in this figure also the share of inorganic and organic forms or different sources. This figure should also include a time serie(s) representing the evolution of the applied physical forcing (e.g. global mean sea surface temp or SAT). In this way, the reader could quickly understand how and what is varied in the simulations and this figure would complement the table and current Fig. 1
  - We agree with the reviewer that such a figure would be useful in case of a transient evolution • of nutrient deposition fields and physical forcing. However, as we stated in Sect. 2, the nutrient inputs to the ocean for PAST, PRESENT, and FUTURE periods are due to an atmospheric simulation using the emission for the years 1850, 2010, 2050, and 2100, respectively. In more detail, the available atmospheric nutrient deposition inputs to the ocean are based on anthropogenic and biomass burning emissions along with the resulted atmospheric chemistry impacts on the gas- and particulate-phases for the single years 1850, 2010, and 2100. Note, however, that a transient simulation of atmospheric tracers from 1850-2100 would require extremely high computational power for the atmospheric chemistry and transport model (which is extremely cost-intensive due to a high number of chemical tracers to be advected and an extremely short time step required for atmospheric models). Therefore, we followed the coordinated CMIP protocol to investigate the effects of atmospheric chemistry AerChemMIP (Collins et al., 2017; https://gmd.copernicus.org/articles/10/585/2017/gmd-10-585-2017.pdf) which recommends only single year emission forcings defined for a limited number of 4 to 6 different year between 1850-2100. Furthermore, compared to the ocean, the atmosphere is comparably well mixed and biomass burning, as an additional source, will be in equilibrium within maximal a couple of months in the atmosphere. The applied physical forcing is kept constant for all simulations of this study, and the whole period (PAST, PRESENT, and FUTURE). This is now clearly stated in Sect. 2.1.1 (see also our reply in GS1).

However, we now provide a new figure in the Supplement that shows the globally averaged atmospheric deposition data simulated by CTM and applied for PISCES. The following text is added in the manuscript (Sect. 2.2): "An example of the globally averaged N, Fe, and P atmospheric deposition data as simulated be CTM and applied in PISCES is presented in Fig. S2. Hence, the here discussed simulations should be considered as idealized sensitivity experiments to estimate the response on the ocean surface biogeochemical properties to changed atmospheric deposition."

Figure S2: Globally averaged atmospheric deposition fluxes (red lines) of a) nitrogen, b) phosphorous, and c) iron in mol  $m^{-2} s^{-1}$ , as simulated by the atmospheric chemistry and transport model and taken into account in PISCES. The black line indicates forcing for the control run under preindustrial conditions (i.e., year 1850).

**II.** Specific Comments**

- SC1. P1, 19: Immediately when reading the abstract one starts to wonder what kind of physical ocean model is used to power PISCES. Please clarify that PISCES is coupled offline (?) to a forced ocean only simulation.
  - We propose to add the following sentence in the abstract: '*PISCES, as part of the EC-Earth model suite, runs here in offline mode using prescribed dynamical fields as simulated by the ocean model NEMO*.'

**SC2. P1, 19: Please mention how atmospheric CO2 and climate change is included.**

• As our study focuses on nutrient cycling and productivity which in NPZD models are independent of atmospheric CO2 (or dissolved inorganic carbon in the water), we think it's better not to mention the atmospheric CO2 mixing ratio in the abstract since it is not a central topic of our study, i.e., the effect on acidity and carbon fluxes is not in the focus here. All the simulations are forced by preindustrial pCO2 and this is now clearly stated in the revised manuscript, both in the model description and the Summary sections, i.e., "Moreover, the atmospheric CO2 mixing ratio is set to the preindustrial value of 284.7 ppm, to effectively isolate the impact of atmospheric deposition on the marine biogeochemistry parameters."

Overall, climate change is not considered here. Our idealistic approach allows us to isolate the effect of atmospheric chemistry and transport changes on productivity undisturbed from any other physical changes and longer-term variability. With climate change included, the conclusion drawn in this study would not be possible or, less robust. However, a detailed discussion on how climate change would affect our results is provided in the new Discussion section now.

**SC3. P3, 129: Is it correct to say that Fe-containing combustion aerosols are mainly deposited in the Pacific and Southern Ocean? Or do you mean that combustion aerosols play a larger role (compared to dust) in these regions?**

- We agree with the reviewer. This part now reads as: 'However, the aerosols from natural and combustion sources tend to be deposited in different regions of the oceans. For example, the subtropical North Atlantic Ocean and the Arabian Sea receive the majority of Fe originated from natural dust aerosols, in contrast to the Pacific and Southern oceans where the Fecontaining combustion aerosols play a more important role compared to atmospheric dust (Hamilton et al., 2020; Ito et al., 2019b).'
- SC4. P5, 119-22: I do not understand what the authors want to say here. Is PISCES now fully coupled online to NEMO and the EC-Earth ESM or rather forced offline with an OMIP simulation? I think it should read: "The state-of-the art biogeochemistry PISCES model is here run "offline" with prescribed transport and T, and S fields (see sec. 2.1). The version of PISCES implemented within NEMO and the European Earth System Model EC-Earth is used in this study. PISCES simulates the ...
  - This part now reads as: "The state-of-the-art biogeochemistry model PISCES (Aumont et al., 2015), enabled here within the framework of the European Community Earth System Model EC-Earth (http://www.ec-earth.org/), is here used in offline modus to investigate the impact of atmospheric deposition fluxes of N, Fe and P on the marine productivity. PISCES (Pelagic Interactions Scheme for Carbon and Ecosystem Studies volume 2), as a part of the Nucleus for European Modelling of the Ocean (NEMO), includes a detailed representation of the lower trophic levels of marine ecosystems."

Moreover, a new section (i.e., Sect. 2.1.1) that describes in detail the forcing data used to run PISCES is now added (please see our reply to SC1).

- SC5. P6, 15: please specify the physical output used to force PISCES.
  - Please see our reply to GC1.

- SC6. P6, l6: which OMIP simulation?
  - Please see our reply to GC1.
- SC7. P6, l6: Please specify how the 1948 to 2009 forcing is aligned in the spin-up of PISCES and the transient simulation from 1850 to 2100.
  - Please see our reply to GC1.
- SC8. P6, 17: I guess this refers to the spin up for ocean model without PISCES? Please clarify.
  Please see our reply to GC1.
- SC9. P6, l8-l10: again to which simulation or model does this initialization apply?
  - Please see our reply to GC1.
- SC10. P6, 18-110: I guess there are still substantial drifts in O2, N, Si, P and Alk after such a short spin up of 300 years only. Please specify how large the drifts are.
  - For this work, we analyzed the results from 1851 to 2100 (namely, PAST: 1851–1870 average, PRESENT: 2001–2020 average, and FUTURE: 2081–2100 average). For the revised version, however, we also performed a control simulation with constant preindustrial deposition inputs (N, Fe, and P) and identical forcing as the STD and ORG transient runs. This new simulation is used for the drift corrections of the model results (please see also our reply to GC2). According to the new control simulation, however, the resulted drifts in surface properties are low and can be considered in equilibrium, without changing our conclusions. Regarding the oceanic concentrations of O2, N, Si, P, and Alk, we present below the relative differences between the drift-corrected and the uncorrected annual mean oceanic concentrations of i) O2, ii) N, iii) Si, iv) P, v) Fe and vi) Alk, for depths from the surface up to ~100m.

---

## Author Comment (AC2) · 31 Aug 2020

We thank the reviewer for the careful reading of the manuscript and all the fruitful comments and suggestions. Please find below our point-by-point replies:

**I. General Comments**

**GC1.** While the authors systematically validated their present day simulation against observations and described the effects of their new atmospheric nutrient deposition fields on surface ocean nutrient concentrations, as well as the rates of primary production and nitrogen fixation, I found that the quantity and organization of the material eclipsed crucial results, and that the depth of the analysis that was presented was somewhat limited. Since the title emphasizes global oceanic productivity, I was expecting significantly more discussion about the emergent rates of primary production and nitrogen fixation (currently ~1 page combined). That biological productivity/nitrogen fixation is relatively stable at the global scale while more significant changes occur regionally implies a compensatory mechanism, which is not really explored. I was looking for more information supported by encompassing and generalizing diagnostics than the numerous supplied maps could provide:

We appreciate the aforementioned valuable and constructive suggestions. Meanwhile, additional simulations were performed, and a deeper analysis of the results allowed more substantial insight into the questions raised by the reviewer. Altogether, this leads to a major revision of the manuscript taking up all the reviewers' suggestions and leading to substantial improvements.

- **How does the ratio of the atmospheric supply of nutrients change regionally/globally?**

  The figure below (middle) shows the N/P ratio of atmospheric deposits relative to the Redfield ratio for the PRESENT and the changes in PAST and FUTURE periods. As can be seen for PRESENT atmospheric fluxes supply, nearly everywhere a surplus of nitrogen compared to phosphorus is demonstrated. By contrast, the same plot for surface water concentrations indicates a deficiency by nitrogen (in agreement with observational evidence). In both FUTURE and PAST periods, the atmospheric N/16P ratio is lowered which would favor N-fixation in warm water regions if not counteracted for by other processes.

  **Atmospheric deposition ratios**

[Figure]

  *Atmospheric nutrient deposition fluxes relative to the Redfield ratio for PRESENT (middle; values >1.0 denotes excess of nitrogen compared to phosphorus) and the relative changes for PAST (left) and FUTURE (right) for the STD simulation.*

**Oceanic Concentration Ratios**

[Figure]

The implications for both productivity and N-fixation and are now discussed in the new Sect. 3.3.1 in the revised version of the manuscript, e.g., "*In regions with significant macronutrient limitations, the elemental ratio of deposited N:P can be, however, important. To estimate the relative impact of the changes in this ratio, we calculated the modeled nitrogen concentrations relative to the model's Redfield ratio (Fig. 5a). For PRESENT, the model exhibits almost everywhere a deficiency with respect to nitrogen (except for some coastal areas). This is in good agreement with data from WOA which likewise indicates a predominant nitrogen deficiency almost everywhere (Fig. 5b). Next, the N:P ratio relative to the Redfield as supplied by atmospheric deposition for PRESENT (middle) together with the changes in PAST and FUTURE (Fig. S8b) is calculated. Hence, a strong excess of N compared to P for modern times is indicated. As a consequence of the model's nitrogen deficiency (Fig. 5a), this atmospheric nitrogen excess maintains higher productivity than without the atmospheric supply. For preindustrial times, the atmospheric N:P ratio is almost everywhere reduced, thus increasing the N-deficiency. Hence, rather the lowered atmospheric nitrogen inputs than the lowered phosphorous inputs in PAST and FUTURE are responsible for the diminished productivity in these experiments. To further demonstrate this, we carried out an additional sensitivity simulation (namely PIP simulation) where, phosphorous atmospheric deposition fluxes kept constant at preindustrial levels, while the other studied atmospheric inputs (i.e., N and Fe) varied as for the STD simulation. As expected, the effect on phosphate concentrations (Fig. S9b) and productivity (Fig. S9d) in this sensitivity simulation remain extraordinarily low, i.e., the relative difference to STD is almost everywhere below 1%. This overall demonstrates that the changes in phosphorus do not play a significant role in marine productivity from preindustrial to future periods.*"

[Figure]

***Figure 5: Molar oceanic ratios N:16P averaged in the upper 20m for PRESENT, as calculated by the model (left) and based on World Ocean Atlas (WOA; Garcia et al., 2010b) data (right). Values >1.0 denote overshoot of N vs P relative to the Redfield ratio (C:N:P =122:16:1); blue areas indicate a surplus of P or deficiency of N.***

Please find a more detailed discussion in the revised manuscript.

- **How does the combination of these resources promote or inhibit production vs diazotrophy?**

The answer is complex since it varies regionally. In PAST most of the emitted nitrogen is deposited in cold water regions where the effect on N-fixation is low, while in some warm water regions iron deposition declines and further limits N-fixation by diazotrophs. In the North Pacific iron seems to be the key driver of found changes with cascading effects involving declining diatom production, nutrient accumulation in the subpolar gyre, and advective nutrient transport to remote regions.

These interesting findings, however, are now detailly discussed in the revised manuscript, i.e., "*PISCES models two phytoplankton functional types: 1) the nanophytoplankton producing calcareous shells and 2) the diatoms producing siliceous shells (Aumont et al., 2015). In the high latitudes, a large part of productivity is related to siliceous diatoms (e.g., Malviya et al., 2016; Uitz et al., 2010) which is accounted for in the model by low nanophytoplankton to diatoms ratios (Fig. 6b). Accordingly, the overwhelming part of productivity reduction in the northernmost Pacific (Fig. 3d) is related to the decline of diatoms. This is well reflected by the increase of the nanophytoplankton to diatoms ratio for PAST relative to PRESENT (Fig. 6a). In turn, this leads to enhanced silicate concentrations in the North Pacific (Fig. 7a). Part of the unutilized silicate is advected southward via the North Pacific Current and the California Current leading also to elevated concentrations along the western coast of North America (Fig. 7a). A further consequence of the strongly diminished productivity is an accumulation of nitrogen in the subpolar gyre of the North Pacific (Fig. 2a). The nitrogen anomaly is strongest in the southwestern area of the gyre and part of the excess nitrogen is injected into the northern California Current. As a result, a strong positive and wedge-shaped productivity anomaly develops in front of western Canada in PAST (see Fig. 3d). The positive anomaly is caused by the increased production of nanophytoplankton productivity (not shown) which dominates in this region (Fig. 6b); north of the wedge lowered iron limits productivity while south of the wedge nitrate is limiting. Altogether, this reflects a slight shift from diatom production to nanophytoplankton in the eastern Pacific north of 40 °N, as indicated by a decline of ~10% of the nanophytoplankton to diatoms concentrations in the upper 20 m (see Fig. 7b).*

*Apart from the northernmost Pacific, the decline in diatom production leads almost everywhere to slightly increased silicate concentrations in PAST (Fig. 7a). Productivity changes in the Southern Ocean remain low (Fig. 3d) for PAST. The reason for this is the strong light limitation around Antarctica (Fig. 4b) and the deep mixed layer which suppresses productivity and subsequently builds up a large pool of unutilized nutrients. Part of the unutilized nutrients is advected further north into the Southern Ocean, driving productivity there. Accordingly, the reduced deposition of nitrogen and iron in this area (Figs. 1a,d) have only a slight impact on productivity. Consequently, this region is relatively robust against external nutrient input maintaining stable productivity. A similar effect is seen for the North Atlantic where vigorous exchange with Arctic waters takes place across the Norwegian-Greenland Sea. By contrast, in the subpolar North Pacific, the import of unutilized nutrients from the Arctic is hampered, as the water exchange with polar waters is limited by the shallow Bering Strait and the Aleutian Arc. Therefore, the North Pacific appears most sensitive to external nutrient inputs.*"

**a)**

[Figure]

**b)**

[Figure]

*Figure 7: a) PAST to PRESENT relative differences (%) of silicate surface oceanic concentrations of as calculated by the model for the STD simulation; b) Seawater concentration ratio of nanophytoplankton to diatoms in the upper 20m, averaged over the NW Pacific (east of 200°E and north of 40°N). The red line indicates primary production rates for the STD simulation and the black line for the CTRL simulation, respectively.*

- **Are phytoplankton (or diazotrophs) consuming critical resources "upstream" that inhibit "downstream" productivity via scarcity or changing nutrient ratios?**

  Indeed, we found indications for this in the mid to high latitude North Pacific, where a decline in iron lowers productivity, which leads to enrichment of nitrogen which is subsequently transported southeastward where these waters cause a positive anomaly in nanophytoplankton. A comprehensive analysis is now given in the revised version (see also our reply to the previous comment), e.g.," *The North Pacific turns out to be most sensitive to iron deposition. For the preindustrial period, the lowered input of iron to this region leads to a strong decline of siliceous diatom production leading to an enrichment of silicate, nitrogen, and phosphorus. In turn, this leads to enhanced equatorward transport of nutrients resulting in elevated production rates of calcareous nanophytoplankton in the south-eastern North Pacific. Overall, the North Pacific appears most sensitive to external nutrient deposition mainly due to two reasons: 1) the strongest deposition changes take place in the northern mid to high latitudes, and 2) that compared to the Southern Ocean and the North Atlantic, the exchange with cold and nutrient-enriched polar waters is limited by land by the shallow Bering Strait and the Aleutian arc. By contrast, the southern high latitude ocean contains a large amount of unutilized nutrients that are advected further north (to mid-latitudes) making this region more robust against changes in external nutrient input. In agreement, however, with observational evidence from WOA, PISCES exhibits a widespread surplus of nitrogen compared to phosphorous and with respect to the Redfield ratio. Therefore, the applied changes in phosphorus inputs have nearly no impact on primary production in the model. Note that this applies even to the warm water regions, where reductions*

*in atmospheric iron supply limit nitrogen fixation by diazotrophs in both PAST and FUTURE periods.*"

- **Are unutilized nutrients (e.g. Southern Ocean iron in the future) transported away from the surface to reemerge elsewhere and stimulate productivity remotely?**

Yes, we found the Southern Ocean (and as well the North Atlantic) to be more robust against changes in atmospheric deposition than the N. Pacific. In the Southern Ocean, extremely cold temperatures around Antarctica (thermal isolation due to the ACC) and the widespread lack of iron further delimit productivity and thus nitrogen-enriched water are advected equatorward and maintain vigorous productivity in mid-latitudes where the iron limitation is of minor importance. The additional Fe around Antarctica in FUTURE is deposited around the coast where strong light limitation exists (we show this in the revised version. Therefore, the effect on productivity is low and nutrients are advected equatorward and maintain more or less stable production in the mid-latitudes.

In the North Pacific injection of nutrient-enriched Arctic waters is effectively suppressed by the shallow Bering Strait and the Aleutian volcanic arc while in the North Atlantic exchange with Arctic waters is maintained by the Norwegian current and East Greenland Current.

Please find for more details in the revised manuscript, e.g., "*This work documents an overall low impact of atmospheric nutrient deposition scenarios on total marine primary production on a global scale. This is because much of modern productivity is driven by nutrients already recycled in the euphotic zone or by nutrient import from the deep ocean (such as in upwelling regions). Additionally, atmospheric transport appears rather important, as a significant part of nutrient deposition takes place in the northern high latitudes, where light conditions and temperature further limit productivity. Accordingly, even substantial reductions of nitrogen, phosphorus, and iron, ranging between 36 and 51% during the preindustrial period result in an only modest decline of primary production of about 3%. However, substantial local productivity changes of up to 20% were found in regions today limited by nutrients. The strongest sensitivity against atmospheric nutrients is found for the oligotrophic subtropical gyres of the North Atlantic and North Pacific. In these regions, good light conditions and warm temperatures together with low nutrient concentrations predominate. Additional atmospheric nutrient input to this region immediately results in production by increasing the biogenic turnover.*

*The North Pacific turns out to be most sensitive to iron deposition. For the preindustrial period, the lowered input of iron to this region leads to a strong decline of siliceous diatom production leading to an enrichment of silicate, nitrogen, and phosphorus. In turn, this leads to enhanced equatorward transport of nutrients resulting in elevated production rates of calcareous nanophytoplankton in the south-eastern North Pacific. Overall, the North Pacific appears most sensitive to external nutrient deposition mainly due to two reasons: 1) the strongest deposition changes take place in the northern mid to high latitudes, and 2) that compared to the Southern Ocean and the North Atlantic, the exchange with cold and nutrient-enriched polar waters is limited by land by the shallow Bering Strait and the Aleutian arc. By contrast, the southern high latitude ocean contains a large amount of unutilized nutrients that are advected further north (to mid-latitudes) making this region more robust against changes in external nutrient input. In agreement, however, with observational evidence from WOA, PISCES exhibits a widespread surplus of nitrogen compared to phosphorous and with respect to the Redfield ratio. Therefore, the applied changes in phosphorus inputs have nearly no impact on primary production in the model. Note that this applies even to the warm water regions, where reductions in atmospheric iron supply limit nitrogen fixation by diazotrophs in both PAST and FUTURE periods.*"

- **Are there teleconnections associated with regions of enhanced export and enriched deep water nutrients upwelling elsewhere?**

Advection by deep waters is generally slow, and significant effects will probably emerge on multi-centennial time scales. Furthermore, diffusive mixing takes place with nutrient-rich deep waters. This makes it difficult to detect in our scenarios. In general, the export of nutrients can take place in regions where light and temperature limit productivity (further constraints relate to iron). The below figure displays exemplary the degree of light limitation for calcareous nanophytoplankton.

[Figure]

Low values indicate the predominance of light limitation. Outside the polar regions where light/temperature dominates changes reflect mainly changes forced by altered productivity due to the self-shading effect. In the North Pacific light limitation clearly declines in PAST due to the more effective iron limitation. This leads to enhanced nitrogen transports southwards in regions with predominant N-limitation and subsequently enhanced productivity there.

In the revised version limitations are discussed in a broader and more comprehensive context including changes in other limiting factors (i.e., nutrients, light, Fe). Please find for more details in the revised manuscript, e.g., " *Despite the relatively strong changes in total atmospheric nutrient supply from PAST to FUTURE (Table 1), the impact of atmospheric nutrients on the global productivity rates remains low in the model. This is not unexpected, however, as the atmospheric nutrient supply constitutes only a small fraction of the total ocean nutrient inventory. In addition, oceanic regions that are not nutrient-limited today are less sensitive to external nutrient supply. Finally, a large part of primary production is regenerated by remineralized nutrients from particulate organic matter (mainly detritus) in the upper ocean layer.*

*To further identify the oceanic regions that are particularly sensitive to changes in external nutrient inputs, the limiting factors for local productivity in the model are investigated (Fig. 4). Note that we here focused primarily on changes in PAST compared to PRESENT because, in most of the cases, the changed depositional fluxes in FUTURE are roughly in the same direction as in PAST (but lower in magnitude). Figure 4a displays limitations due to nitrogen or phosphorus. High values indicating low limitation are seen in regions that are subject to intense upwelling, like in the equatorial divergence zones or the western margins of NW and southern Africa and South America (coastal upwelling). Accordingly, these regions are less sensitive to atmospheric deposition as nutrients are supplied from deep ocean layers. Lower nutrient limitation is likewise seen in the mid to high latitudes where limitations by temperature and light (Fig. 4b) limit the growth rates. Exceptions are the North Pacific, the Southern Ocean, and the equatorial Pacific where iron limitation matters (Fig. 4c). Consequently, the model's nutrient sensitivity is largest in the subtropics, in particular in the subtropical gyres where good light conditions and warm waters support high growth rates paralleled by diminished nutrient supply from depth due to Ekman pumping. Furthermore, these regions are far from land nutrient sources and so, a major part of total primary production relates to regenerated production (not directly forced by external nutrient supply). This makes the subtropical gyres sensitive to changes in the external atmospheric nutrient.*"

Please find a more detailed discussion in the revised manuscript.

- **What about silicate (Si) fluxes?**

Si atmospheric deposition fluxes into the ocean do not change from PAST to FUTURE since they are solely based on present-day dust deposition fields, as simulated from the CTM for the year 2010 (see also our reply in SC8). The atmospheric Si inputs are calculated by assuming a constant fraction of 30.8% Si by weight in the dust. 7.5% of the deposited Si is assumed soluble and thus entered in the dissolved silicate pool of the model upon deposition.

[Figure]

Simulated changes in PAST and PRESENT oceanic concentrations are therefore related to changed productivity patterns (mainly diatoms). The figures below confirm that due to the lowered productivity in PAST, Si concentrations increase almost everywhere, but stronger in the N-Pacific, and further transport out of the subpolar gyre along the California Current (see below left).

[Figure]

- **How did the composition of phytoplankton functional groups (diatoms vs other phytoplankton) change?**

As an example, we here compare the ratio of nanophytoplankton to diatoms concentration with the iron limitation term in the respective experiments. The above figure shows the ratio of nanoplankton: diatoms (middle) and changes in PAST and FUTURE experiments. The pattern is strongly determined by iron limitation with the strongest impact in the North Pacific where iron and silicate consuming diatoms are diminished compared to nanophytoplankton.

[Figure]

This likewise explains the higher silicate concentrations in the PAST. The weaker response in the FUTURE with partly decreased iron limitation is related to higher iron

inputs in this region compared to PAST and PRESENT. These figures and the respective discussion are now included in the revised manuscript. The following text is added in the revised manuscript: "In the high latitudes, a large part of productivity is related to siliceous diatoms (e.g., Malviya et al., 2016; Uitz et al., 2010) which is accounted for in the model by low nanophytoplankton to diatoms ratios (Fig. 6b). Accordingly, the overwhelming part of productivity reduction in the northernmost Pacific is related to the decline of diatoms. This is well reflected by the increase of the nanophytoplankton to diatoms ratio for PAST relative to PRESENT (Fig. 6a). In turn, this leads to enhanced silicate concentrations in the North Pacific.

[Figure]

*Figure 6: Nanophytoplankton to diatoms oceanic concentrations ratio averaged in the upper 100m for PRESENT (middle), and relative changes for PAST (left) and FUTURE (right) for the STD simulation.*

- **Is production limited by a top-down grazing pressure, or a bottom-up resource limitation?**

Both are true. Grazing by zooplankton delimits phytoplankton production and is most effective during intense blooms. In turn, in oligotrophic regions, the lack of nutrients limits production as well.

- **Some of these issues were touched upon when explaining the counterintuitive higher oceanic P concentrations simulated for the preindustrial era despite lower P deposition, which I found really interesting.**

Indeed, our simulations demonstrate that the increase in P deposition fluxes into the global ocean from PAST to PRESENT is of minor importance for oceanic productivity. As a result, the present-day phosphate oceanic concentrations are calculated lower, compared to preindustrial times. To further demonstrate this, we performed an additional simulation as for STD, but keeping the DP deposition inputs to preindustrial (PI) levels (namely PIP). As expected, we get almost identical present-day oceanic phosphate concentrations and for primary productions as well. This indicates that marine biogeochemistry is more important in controlling phosphorus concentrations at the surface ocean than the direct atmospheric deposition of phosphorus, as we stated in the manuscript. The following text is added in the revised manuscript: "*As expected, the effect on phosphate concentrations (Fig. S9b) and productivity (Fig. S9d) in this sensitivity simulation remain extraordinarily low, i.e., the relative difference to STD is almost everywhere below ~1%. This overall demonstrates that the changes in phosphorus do not play a significant role in marine productivity from preindustrial to future periods.*"

[Figure]

*Figure S1: Surface oceanic concentrations (mmol m$^{-3}$) of phosphate (top row), primary production rates (kg-C m$^{-2}$ yr$^{-1}$) (middle row), and nitrogen fixation (kg-N m$^{-2}$ yr$^{-1}$) (bottom row), as calculated by the model for PRESENT for the sensitivity PIP simulation (i.e., as for STD, but keeping phosphorus atmospheric deposition to preindustrial levels) (left column), and the respective relative differences (%) to the STD simulation (right column).*

**GC2.** **There are many "moving parts" associated with this study that some idealized experiments might help disentangle the mechanistic role of atmospheric nutrient deposition on ocean biogeochemistry and production. Perhaps substitution experiments with the newly derived N, P, or Fe deposition singly swapped with remaining "standard" PISCES inputs (or combinations of two substituted out of three).**

- We agree that more sensitivity experiments would be beneficial for a deeper analysis. This approach is limited, however, by the available resources. Nevertheless, we carried out an additional sensitivity simulation where phosphorous deposition fluxes kept constant at preindustrial levels while all other atmospheric inputs (i.e., N and Fe) changed. This run allows new insight on the importance of N and P macronutrients which is presented in the revised version. Please see our overall reply to SC1.

**GC3.** **I appreciated the signposting of the manuscript structure at the end of the introduction, but I thought the paper would benefit from having separate "results" and "discussion" sections, with integrative diagnostics in the former, and more emphasis on explaining the changes in emergent ocean properties and comparisons with previous studies such as Krishnamurthy in the latter. At the moment, the key messages are very much buried within the qualitative/semi-quantitative description of the results.**

- We agree with the reviewer and for this, the manuscript has undergone a major revision, including now a more detailed analysis. As suggested, we separated the discussion from the results in the revised version of the manuscript. For this, the discussion of productivity has completely rewritten and is substantially expanded now. Respectively, in the revised version, the conclusions are rewritten more concisely.

**GC4.** **I also think that the model-data comparisons, although reassuring, interrupted the flow of presentation. One could create a new section on model validation, but I would recommend moving the material and figures to Supplementary information.**

- Indeed, our aim was not to repeat the work of previous studies but to show, in short, that our version of the model reproduces the main features from the observations. Thus, a separate section of a model evaluation would be very limited and out of the scope of this work. However, we agree with the reviewer that the model-evaluation figures may interrupt the presentation of this work and for this, the model evaluation part has now moved to the Supplementary material of the revised version, as suggested.

**GC5.** **One comparison that may have been really interesting to include is a comparison of the "CTRL" run to a vanilla PISCES simulation, with the standard atmospheric nutrient fluxes.**

- A comparison with the standard atmospheric inputs in PISCES would be of course interesting, but we believe that the value of this study lays on providing new data set of atmospheric deposition inputs for the preindustrial, the present, and future based on realistic atmospheric chemistry and physics (i.e., based on a state-of-the-art atmospheric transport and chemistry model) and plausible scenarios for the future and past. For the standard PISCES configuration, the Fe and P deposition inputs to the ocean are based on the same dust deposition file and based on constant nutrient content and solubilities on the deposited dust aerosols. In contrast, for this work, we provide deposition fluxes based on a detailed mineralogy dataset, online mineral dissolution processes, and atmospheric chemistry for three periods (i.e., namely the years 1850, 2010, and 2100). Such a comparison would be meaningful only for the present-day deposition fields. However, for this work, we intend to rather focus on differences between different periods. Nevertheless, in the case of such a comparison, we do not expect fundamental differences at least on a global scale, as also denoted by current model evaluation compared to previous studies. Though, regional changes could be more important. All in all, we did not provide a comparison with a standard input simulation in this work, since this would be out of the scope of this work.

**II. Specific Comments**

**SC1.** **P3, line 20: "no-linearly" typo.**
- Changed to "*non-linearly*"

**SC2.** **P4, line 13: "It has also been suggested. . ." citation needed, unless it's Krishnamurthy et al. (2010) in which case rephrase for clarity.**
- This part is rephrased as "*Krishnamurthy et al. (2010) also suggested that the simultaneous anthropogenic N and Fe deposition can increase oceanic productivity by 1.5 Pg-C yr-1, corresponding overall to a reduction of atmospheric pCO2 level by ~2.2 ppm by the year 2100.*"

**SC3.** **P6, line 7: "five iterations" I think it would be more precise to say you ran the model for 300 years, repeating the 60 year physical forcing five times. Five iterations could technically imply a spin-up of 5x2700s.**
- We agree with the reviewer that this statement may be confusing. We rewrote this part by adding a new subsection (i.e., Sect. 2.1); please see also our reply to GC1 of Reviewer #1.

**SC4.** **P6, line 9: Which versions of WOA and GLODAP did you use (if not WOA2013 and GLODAPv2). How was DIC initialized? Also from GLODAP?**
- This part is now read as: "*For the initialization of the ocean biogeochemical fields, the climatological fields of oxygen, nitrate, silicate, and phosphate from the World Ocean Atlas 2009 (WOA; Garcia et al., 2010a, 2010b) along with dissolved inorganic carbon (DIC) and alkalinity from the Global Ocean Data Analysis Project (GLODAP; Key et al., 2004) were adopted.*"

**SC5.** **P6, line 25: "no extra optimizations for the iron scavenging parameters" The specific iron cycle configuration is of critical importance to understanding the effect of changing iron input, please can you give more details about this? Did you use particle dependent scavenging? How is organic ligand complexation parameterized (constant or variable ligand concentration)?**
- We agree with the reviewer that the iron cycle configuration is of high importance for Fe and other nutrients oceanic concentrations. Since we are mostly focused here on the differences on productivity and nutrients oceanic concentrations solely due to different atmospheric input parameterizations, the simple chemistry model of PISCES (i.e., based on one ligand (L) dissolved inorganic Fe and dissolved complexed iron (FeL)) is just used and not, for example, the complex chemistry scheme that is also available in the model (i.e., ln_fechem = F), as developed by Tagliabue and Arrigo (2006) and Tagliabue and Völker (2011), which is based on five iron species and two ligands and better match with observations. The ligand concentration in the ocean for this work is kept constant, equal to 0.6 nmol $L^{-1}$ and the scavenging rate by dust is equal to 150 $d^{-1}$ $mg^{-1}$ L (see Table 1 in Aumont et al., 2015).
  We propose to add the following part in the manuscript: "*For this work, the simple chemistry scheme based on one ligand (L) dissolved inorganic Fe and dissolved complexed iron (FeL) is used. Additionally, the ligand concentration in the ocean, is kept constant, equal to 0.6 nmol $L^{-1}$ and the scavenging rate by dust is equal to 150 $d^{-1}$ $mg^{-1}$ L (see Aumont et al., 2015 and ref. therein).*"

**SC6.** **P7, lines 1-12: Timeseries of the model nutrient sources would clarify how the experiment was run i.e. I think you did one transient run 1651-2100 and analyze the nutrient concentrations/ecosystem response at three 20-year average periods. In addition, it would be great to show the temporal evolution of globally/regionally-averaged nutrient concentrations and emergent diagnostics during this run. This got me thinking about**

**whether the "present day" actually represents the peak in atmospheric nutrient deposition, or if that was earlier (70's, 80's or 90's)? There was no real justification for choosing the 2001-2020 average given.**

- The revised manuscript (supplement) contains now time series with globally averaged depositional inputs for P, N, and Fe which clarifies how the model was forced. We also reformulated the description of the experiment in the text. The time series clearly demonstrates that the period 2000-2020 is the one with the highest deposition fluxes into the ocean, i.e.,

[Figure]

*Figure S2: Globally averaged atmospheric deposition fluxes (red lines) of a) nitrogen, b) phosphorous, and c) iron in mol m$^{-2}$ s$^{-1}$, as taken into account in PISCES. The black line indicates forcing for the control run under preindustrial conditions (i.e., year 1850).*

- Indeed, we here performed simulations from 1651-2100, using the first 200 yrs. (i.e., 1651-1850) as a spin-up period for our experiments. For the analysis of the results, we used three 20-year average periods, corresponding to PAST, PRESENT, and FUTURE periods. The simulations with the atmospheric chemistry model comprise yearly simulations for years 1850, 2010, and 2050/2100 (using ACCMIP emissions from Lamarque et al, 2013). The reason is that the atmospheric CTMs are among the most expensive models, so the performance of transient runs over multi centennials is practically impossible. However, considering the typical residence time of tropospheric aerosols of only a few days a 1-year simulation is by far sufficient to bring the atmosphere in equilibrium. Note that for all CTM simulations a one-year spin-up is performed. Afterward, for the ocean depositional forcing, a linear interpolation between the years of the atmospheric run is applied. A more detailed explanation is now provided in the revised version, in the new Sect. 2.2. i.e.,

  *"Simulations with the atmospheric transport and chemistry model are, nevertheless, extremely expensive. Therefore, limitations in available computational resources made it necessary to reduce the CTM simulations to representative single years for 1) the preindustrial state (before 1850), 2) the present-day state (representing the year 2010), and 3) a mid-century (2050), as well as, an end of century (2100) state. However, as the typical residence time of tropospheric aerosols is in the order of a few days, the atmospheric depositional fields used in PISCES represent a well equilibrated atmospheric chemistry and deposition flux, without the need of time transient simulations.*

*For the ocean biogeochemistry model spin up (i.e., from 1651 to 1850) the preindustrial field (the year 1850) was applied. After the 200 years spin-up period, the atmospheric deposition input data for the STD and ORG simulations were linearly interpolated from preindustrial to present-day conditions (i.e., the year 2010) to smoothly capture the transition from past to the modern conditions (e.g., Krishnamurthy et al., 2009). Respectively, the deposition data from the present day were linearly interpolated to the projected estimates (i.e., the years 2050 and 2100). Note that for all temporal and spatial interpolations of this work, as well as for the drift calculations applied for this work, the Climate Data Operators (CDO v.1.9.8) software, as provided by the Max Planck Institute for Meteorology, is here used (https://code.mpimet.mpg.de/projects/cdo/embedded/cdo.pdf; last access 29/02/2020). An example of the globally averaged N, Fe, and P atmospheric deposition data as simulated by the CTM and applied in PISCES is presented in Fig. S2. Overall, the here discussed simulations should be considered as idealized sensitivity experiments to estimate the response on the ocean surface properties to changed atmospheric deposition."*

**SC7.     P7, line 12: Will these datasets be available online for ESM groups to experiment with?**

- We thank the reviewer for this comment. The atmospheric deposition datasets used for this study (past, present, and future) will be available in Zenodo. A relevant statement is added to the Data availability section at the end of the manuscript.

**SC8.     P11, line 18: How do dust and aerosol emissions, that are not considered, vary over the time period in question? I think this is touched upon in the "summary".**

- The deposition data we used for this study come from a CTM simulation using anthropogenic and biomass burning emissions (gases and aerosols) for the past (1850), present (2010), and future projected (2050/2100) eras. However, due to the nature of CTMs (i.e., offline models), these simulations do not consider changes of the meteorology for the preindustrial era or projected meteorology. Thus, dust emissions that are wind-driven, besides the impact of land-use changes, do not vary in these simulations either. Overall, changes in the deposition fields applied in PISCES for this study, represent changes in nutrient concentrations due to anthropogenic and biomass burning emissions as well as the respective impact of atmospheric chemistry (i.e., atmospheric processing). To make more clear, the following part is now added in the new discussion section (i.e., now Sect. 5): "*All changes in nutrient deposition fluxes here accounted for are solely driven by changes in the anthropogenic and biomass burning emissions, along with the changes in insoluble to soluble conversions rates due to atmospheric processing. Thus, the atmospheric deposition fields used in this study did not account for any changes in dust and bioaerosol emissions. Instead, they were kept constant to the present-day atmosphere (i.e., the year 2010), although several studies suggest that dust fluxes may be sensitive to climate change and the land-use changes (e.g., Ginoux et al., 2012; Mahowald et al., 2010; Prospero and Lamb, 2003), and thus could be an important driver of the atmospheric nutrient cycles.*"

**SC9.     P11, line 26: "cooler water temperatures. . ." caused also by high latitude upwelling?**

- The main effect is the cooler mean climate in the high latitudes. The imprint upwelling (N-Pacific) or deep (and convective) mixing (e.g. Labrador Sea North Atlantic) also affects the pattern of SST in the model.

**SC10.     P12, line 5: "leads to more efficient export." N supply may certainly lead to increased rates of export in nitrate-limited oligotrophic regions, but if the nutrients are**

**drawn down to the same low levels, for example in the gyres, is the export actually more efficient?**

- We agree with the reviewer. We replaced the word "efficient" with "*increased*".

**SC11.        P12, line 33: It would be an even more convincing model-data comparison if the authors took advantage of the extensive GEOTRACES iron dataset (https://www.geotraces.org/geotraces-intermediate-data-product-2017/) with 6 years of cross calibrated additional data from a concerted international effort.**

- For this work, we only chose the previous GEOTRACES version because it facilitates comparison with previous studies (e.g., Aumont et al. 2015). As also stated in GC4, we use here the GEOTRACES dataset to demonstrate that our simulations produce realistic oceanic concentrations compared to previous studies, and for this, in the revised version we moved the model evaluation in the supplement (please see our reply in GC4).

**SC12.        P13, line 21-25: side note about Redfield ratios might be better placed with the model set up.**

- The following sentence has been added to the new Sect. 2.1 (i.e., model set-up): "*The model simulates the biogeochemical cycles of carbon and the main nutrients (N, P, Fe, and Si) and includes external nutrient sources from atmospheric deposition, rivers, sea ice, sediment dissolution, and hydrothermal vents, and a constant Redfield ratio (i.e., C:N:P = 122:16:1) for growth of phytoplankton*".

*SC13.*        **P14, line 14: Why does nitrogen fixation decrease?**

- We agree with the reviewer that this should be better explained. For this, we have analyzed it in more detail in the revised manuscript and further relate it to the decreased iron concentration in the PAST (and the FUTURE experiments), i.e., "*Note, however, that nitrogen fixation in PISCES is restricted to warm waters (i.e., above 20 °C). Therefore, the strong reductions of nitrogen deposition in the mid to high latitude North Pacific in PAST have no direct impact on nitrogen fixation. In the subtropical Pacific reduced nitrogen fixation rates mainly reflect the diminished iron input (Fig. 1d). On a global scale, the model calculates overall only a small decrease (~0.2%; Table 1) in preindustrial nitrogen fixation rates compared to present-day, mainly as a result of the decreased soluble iron inputs in the subtropical North Pacific (Fig. 1d). For the future conditions, the model likewise calculates a modest decrease in the global nitrogen fixation (~1%; Table 1) along with decreased iron inputs to the ocean (Figs. 1c,f, respectively) resulting overall in some lower rates of up to 10% in the Equatorial Pacific Ocean (Fig. 3c).*"

**SC14.        P14, line 22: "with the projected decrease of the global inorganic nitrogen and iron inputs. . ." Nitrogen fixation should be promoted by lower N:P ratio (i.e. decreased N and increased P) so is the lower fixation rate due to iron limitation? Is it possible to show maps of resource limitation from the model for phytoplankton/diazotrophs, e.g. the limiting terms in Equation 6 in Aumont et al. (2015)?**

- This is true; in PAST simulation, nitrogen concentrations slightly increase or do not change while phosphate increases strongly. Only iron decreases, overall demonstrating the importance of iron deposition. We have unfortunately not outputted the limitation term for diazotrophs (only for nanophytoplankton and diatoms).

**SC15.        P15, line 32: "all dissolved organic matter is assumed to be instantaneously remineralized. . ." I think this is incorrect. Equation 32 in Aumont et al. (2015) shows how dissolved inorganic matter (for carbon and other species related by fixed Redfield ratios) is separately modeled as a pool supplied by phytoplankton and zooplankton exudation and remineralized aerobically or anaerobically by bacteria over a timescale of the order of months to years.**

- We thank the reviewer for attracting our attention to this issue. We rephrased this part as: "*Note that as for the riverine organic fractions in the model (see Aumont et al., 2015), we assume here an instant transformation of the atmospheric dissolved organic nitrogen (DON) and organic phosphorus (DOP) inputs to the respective inorganic fractions in the water column.*"

**SC16.** **P16, line 19: "in contrast to the rather balanced nitrogen fixation rates. . ." a 2% change in primary production also sounds rather balanced to me.**
- We agree with the reviewer and we removed this part.

**SC17.** **P16, line 20-28: 15-20% increases occur relatively widely in the ocean, so what causes the counterbalancing decline in productivity? Why are the decreases confined to these bands in the Pacific?**
- In ORG increased iron input in the sub-polar gyre increases diatom production leading to more consumption of nitrate. Subsequent transport of nitrogen diminished waters further to the south cause a decrease in productivity further south. In the tropical north Pacific, we find a small zonal dipole pattern of decreased (north) and increased (south). The boundary between decreased and increased bands matches the sharp boundary from iron limitation to non-iron limitation (see example below). We suppose increased iron input south (where iron limitation is) stimulates the production and diminishes nitrogen. Advective mixing of the N-diminished waters with water further north decreases productivity leading to the dipole pattern as presented in the old Figure 4 (i.e., in the submitted version).

  This analysis is now included in the revised version, i.e., "*Primary production increases almost in all ocean basins for the ORG simulation (Fig. 8d), except some parts of the Subpolar Pacific Ocean. In particular, higher rates are calculated in the subpolar Atlantic Ocean (up to 15%). In the N-limited oceanic regions, the increased ORG atmospheric nitrogen deposition (Fig. S3b) directly increases the production rates (Fig. 8d). Such a case is the western subtropical North Pacific, where atmospheric N deposition supports an extra production of up to 15%. The production rates are also increased in the subtropical South Pacific and Atlantic Oceans up to nearly 20% in the ORG simulation. In total, the primary production increased from ~46.7 Pg-C yr-1 for the STD to ~47.8 Pg-C yr-1 for the ORG (Table 1). Figure 8d points out to regions in the Pacific where production decreased. For the North Pacific, however, this represents the same mechanism as described above for the differences in primary production rates between PAST and PRESENT. For the ORG simulation, the increased iron input in the Pacific subpolar gyre increases diatom production leading to higher consumption of nitrate (Fig. S10b). Subsequent transport of nitrogen diminished waters further to the south cause a decrease in productivity further south. The boundary between decreased and increased bands matches the sharp transition from iron limitation to nitrogen limitation (Figs. 4a,c). Increased iron input south of the boundary (i.e., where iron limits) stimulates the production and diminishes nitrogen. However, the advective mixing of the N-diminished waters with waters further north decreases productivity north of the boundary (i.e., where N limits). Overall, the result is the dipole pattern as demonstrated in Fig. 8d.*"

[Figure]

*Figure 8: Nitrogen fixation (kg-N m⁻² yr⁻¹) and primary production (kg-C m⁻² yr⁻¹) rates as calculated by the model (a,c) for the ORG simulation for PRESENT (2001–2020 average), and the respective relative (%) differences (b,d) compared to the STD simulation.*

[Figure]

*Figure S3: Oceanic concentrations averaged over the upper 100m (left column) of nitrate (a), iron (c) and phosphate (e) as calculated by the model for the ORG simulation for PRESENT (2001–2020 average), and the respective percentage differences (b,d,f) compared to the STD simulation (right column).*

**SC18.** **P17, line 13-20: Salinity restoring and mixed layer dynamics was never mentioned in the main text, so surprised to see it prominently in the "Summary" section.**

- For this work salinity restoring was only applied during the OMIP run from which the physical ocean forcing for the offline PISCES runs were generated. The salinity of the biogeochemical offline runs is, however, constant, representing only the yearly cycle on a daily basis. To avoid any confusion, we removed this part from the text, since the prolongation of the run for the RCP8.5 scenario is not that relevant for this study. Note, however, that a complete description of the forcing data used for this study is now added in Sect. 2.1 in the revised version.

**SC19.** **Figures: It would be preferable to use a perceptually uniform color palette for the CTRL maps, as opposed to the rainbow/jet colormap currently shown (see here for details: https://blogs.egu.eu/divisions/gd/2017/08/23/the-rainbow-colour-map/, not to mention the accessibility issue surrounding red/green vision deficiency).**

- We thank the reviewer for attracting our attention to this issue. We replotted all figures using standard perceptually uniform colormaps, such as Viridis, (see https://bids.github.io/colormap/)

---

## Author Response (AR2)

We thank the editor for the careful reading of the manuscript and the fruitful comments. Please find below our point-by-point replies:

**I.   Comments to the Author**

1.  **The authors have made extensive changes as a result of the reviewers' comments, and I believe the paper is now suitable for publication. They have enlarged and clarified the description of the model, and completely rewritten the results section (section 3) and much of the summary and conclusions (section 6), while the discussion section (section 5) has been enhanced. The paper does need a good copy edit, as there are many examples where the authors' use of English should be corrected.**
    - We thank the editor for appreciating our efforts to improve our manuscript based on the fruitful comments of the reviewers. As the editor pointed-out, we tried to further improve our use of English, to remove repetitions, and to correct typos.

2.  **In section 2.1.2, they state: "The default N, Fe and P atmospheric nutrient inputs of PISCES in the EC-Earth configuration have been replaced for this work by state-of-the-art monthly mean atmospheric deposition fields recently published in the literature." Surely this needs a reference – should it be Duce et al. 2008?.**
    - We thank the reviewer for attracting our attention to this issue. Indeed, this sentence is not really needed, since we now explicitly present the new deposition fluxes in Sect. 2.2.1 as we clearly state after some lines. i.e., *"In contrast to previous studies (e.g., Aumont et al., 2015), the new N, Fe, and P atmospheric deposition fields considered here (see Sect. 2.2) are all calculated based on…"*. Thus, to avoid any further confusion, we deleted this sentence.

3.  **In section 2.2.2 there is the statement "The highest annual mean DFe deposition fluxes into the ocean (Fig. 3c)…." This should presumably refer to either Fig 1e or Fig. S3c, not 3c, which is about changes in N fixation.**
    - Indeed, this statement refers to Fig. 1e, as clearly stated at the beginning of the paragraph. To avoid repetition, however, we removed this extra reference to the (same) figure (i.e., Fig. 1e).

4.  **In section 3.1 on Fe deposition, I believe the final figure reference should be to S5, not S4.**
    - Corrected.

5.  **In section 3.3.1, there is the sentence "Consequently, the model's nutrient sensitivity is largest in the subtropics, in particular in the subtropical gyres where good light conditions and warm waters support high growth rates, paralleled by diminished nutrient supply from depth due to Ekman pumping." As written, this implies that the diminished nutrient supply is caused by Ekman pumping, whereas in fact Ekman pumping normally increases nutrient supply to the surface. This sentence could perhaps be rewritten as "… support high growth rates, despite the diminished nutrient supply from deeper layers."**

[revised manuscript text omitted]